# Synthesis and Biological Evaluation of Highly Active 7-Anilino Triazolopyrimidines as Potent Antimicrotubule Agents

**DOI:** 10.3390/pharmaceutics14061191

**Published:** 2022-06-02

**Authors:** Paola Oliva, Romeo Romagnoli, Barbara Cacciari, Stefano Manfredini, Chiara Padroni, Andrea Brancale, Salvatore Ferla, Ernest Hamel, Diana Corallo, Sanja Aveic, Noemi Milan, Elena Mariotto, Giampietro Viola, Roberta Bortolozzi

**Affiliations:** 1Department of Chemical, Pharmaceutical and Agricultural Sciences, University of Ferrara, 44121 Ferrara, Italy; lvopla@unife.it (P.O.); bbc@unife.it (B.C.); 2Department of Life Sciences and Biotechnology, University of Ferrara, 44121 Ferrara, Italy; mv9@unife.it; 3Medicinal Chemistry Department, Integrated Drug Discovery, Aptuit—An Evotec Company, Via A. Fleming, 37135 Verona, Italy; chiara.padroni@gmail.com; 4School of Pharmacy and Pharmaceutical Sciences, Cardiff University, King Edward VII Avenue, Cardiff CF10 3NB, UK; brancalea@cardiff.ac.uk; 5Swansea University Medical School, Institute of Life Sciences 2, Swansea University, Swansea SA2 8PP, UK; salvatore.ferla@swansea.ac.uk; 6Molecular Pharmacology Branch, Developmental Therapeutics Program, Division of Cancer Treatment and Diagnosis, Frederick National Laboratory for Cancer Research, National Cancer Institute, National Institutes of Health, Frederick, MD 21702, USA; hamele@dc37a.nci.nih.gov; 7Laboratory of Target Discovery and Biology of Neuroblastoma, Istituto di Ricerca Pediatrica (IRP), Fondazione Città della Speranza, Padova, Corso Stati Uniti 4, 35128 Padova, Italy; d.corallo@irpcds.org (D.C.); s.aveic@irpcds.org (S.A.); 8Hemato-Oncology Lab, Department of Woman’s and Child’s Health, University of Padova, 35131 Padova, Italy; noemi.milan@studenti.unipd.it (N.M.); mariottoelena@gmail.com (E.M.); roberta.bortolozzi@unipd.it (R.B.); 9Laboratory of Experimental Pharmacology, Istituto di Ricerca Pediatrica (IRP), Fondazione Città della Speranza, Padova, Corso Stati Uniti 4, 35128 Padova, Italy

**Keywords:** antimitotic agents, structure–activity relationship, [1,2,4]triazolo [1,5-*a*]pyrimidine, antiproliferative activity, tubulin polymerization

## Abstract

Two different series of fifty-two compounds, based on 3′,4′,5′-trimethoxyaniline (**7a–ad**) and variably substituted anilines (**8a–v**) at the 7-position of the 2-substituted-[1,2,4]triazolo [1,5-*a*]pyrimidine nucleus, had moderate to potent antiproliferative activity against A549, MDA-MB-231, HeLa, HT-29 and Jurkat cancer cell lines. All derivatives with a common 3-phenylpropylamino moiety at the 2-position of the triazolopyrimidine scaffold and different halogen-substituted anilines at its 7-position, corresponding to 4′-fluoroaniline (**8q**), 4′-fluoro-3′-chloroaniline (**8r**), 4′-chloroaniline (**8s**) and 4′-bromoaniline (**8u**), displayed the greatest antiproliferative activity with mean IC_50′_s of 83, 101, 91 and 83 nM, respectively. These four compounds inhibited tubulin polymerization about 2-fold more potently than combretastatin A-4 (CA-4), and their activities as inhibitors of [^3^H]colchicine binding to tubulin were similar to that of CA-4. These data underlined that the 3′,4′,5′-trimethoxyanilino moiety at the 7-position of the [1,2,4]triazolo [1,5-*a*]pyrimidine system, which characterized compounds **7a–ad**, was not essential for maintaining potent antiproliferative and antitubulin activities. Compounds **8q** and **8r** had high selectivity against cancer cells, and their interaction with tubulin led to the accumulation of HeLa cells in the G2/M phase of the cell cycle and to apoptotic cell death through the mitochondrial pathway. Finally, compound **8q** significantly inhibited HeLa cell growth in zebrafish embryos.

## 1. Introduction

The 2016 World Health Organization update, the most recent available, listed the top three causes of disease mortality as ischemic heart disease, cancer and stroke [1,2]. In 2020, 19.3 million new cases of cancer were diagnosed according to the International Agency for Research on Cancer, with 10.0 million deaths [3]. In the next four decades, cancer deaths are expected to be the primary cause of death, with a 2.08-fold increase by the year 2060 [4] and the leading cause of mortality after 2030 [5]. For these reasons, it is very important to develop drugs that specifically act on malignant cells, ideally through use of pathways limited to these cells.

Amongst cancer therapeutic targets, drugs targeting microtubules represent one of the most effective classes of cancer chemotherapeutic compounds available to date [6]: these act as disruptors of mitotic spindle assembly, and, for this reason, are called antimitotic agents. Cells treated with these compounds fail to complete cell division and undergo an apoptotic death process [7].

Microtubules are composed of αβ-tubulin heterodimers that associate to form hollow cylindrical structures, and they are highly dynamic structures that, together with microfilaments and intermediate filaments, form the cellular cytoskeleton [8]. Besides its prominent role in mitosis, the microtubule network is recognized for its role in regulating cell growth and movement and key signaling events that modulate fundamental cellular processes [9]. The dynamicity of microtubules results in their constant lengthening and shortening throughout all phases of the cell cycle [10].

Microtubule-targeting agents (MTAs) are compounds that bind to tubulin [11]. They disrupt microtubule dynamics in distinct ways, and they are classified into two main groups: destabilizing and stabilizing agents [12]. The end action of the two classes of drugs is suppression of microtubule dynamics [13]. At relatively high concentrations in biochemical assays with tubulin, MTAs either inhibit microtubule polymerization, destabilizing microtubules and decreasing microtubule polymer mass, or promote microtubule polymerization, stabilizing microtubules and increasing polymer mass [14].

The colchicine site of tubulin is one of the most important pockets of the protein, and it has been a focus for the design of tubulin-destabilizing agents [15,16,17]. Numerous natural and synthetic compounds that bind to tubulin at the colchicine site have been identified [18,19]. One of the most important is combretastatin A-4 (CA-4, 1a, Figure 1). Isolated from the South African tree *Combretum caffrum* Kuntze [20], CA-4 has remarkable cytotoxic effects in a wide variety of preclinical tumor models [21] and inhibits tubulin polymerization by interacting at the colchicine site [22]. Although CA-4 is a promising clinical candidate, with very low associated side effects, it is not an ideal drug. It has very poor aqueous solubility, a short biological half-life and undergoes *cis*-*trans* isomerization in heat, light and protic media, forming the totally inactive *trans* isomer from the active *cis*-form [23]. To improve the solubility of CA-4, its phosphate salt (CA-4P, 1b) was synthesized, and this compound is still in clinical development for the treatment of ovarian and other cancers [24,25].

The [1,2,4]triazolo [1,5-*a*]pyrimidine molecular skeleton was identified as a versatile pharmacophore for the development of a limited number of chemically diverse small molecules identified as microtubule active compounds that, depending on their substitution pattern, either promote microtubule stabilization or disrupt microtubule integrity [26,27,28,29]. Compound **2a** and its des-methyl analogue cevipabulin (TTI-237, **2b**) competed with vincristine at the binding site of vinca alkaloids, interacting with tubulin heterodimers, thereby leading to an overall disruption of microtubule integrity [26,27,28,29]. Congener **3**, when compared to **2b** structurally, features a relatively bulky bicyclic amine substituent but lacks the alkoxy side chain of the fluorinated phenyl ring at the 7- and 6-positions, respectively, of the triazolopyrimidine core. Compound **3**, however, binds exclusively to microtubules and not to unpolymerized tubulin heterodimers, promoting microtubule-assembly and stabilization [29].

Because it is well known that the 3,4,5-trimethoxyphenyl ring is a characteristic structural requirement to maximize activity in a large series of inhibitors of tubulin polymerization based on CA-4 pharmacophores, more recently two different classes of regioisomeric triazolopyrimidines that retain the 3,4,5-trimethoxyphenyl ring at the 2- or 7-position of the triazolopyrimidine core were synthesized [30,31].

Yang et al. described derivatives possessing a common 3,4,5-trimethoxyphenyl group at the 2-position of the 2-methyl-5-acetyl-[1,2,4]triazolo [1,5-*a*]pyrimidine nucleus and different substituted phenyl moieties at its 7-position, with compounds **4** and **5** as the more promising members of the series [30]. Derivative **4** inhibited the growth of the HeLa and A549 cell lines with IC_50_ values of 0.75 and 1.02 μM, respectively, but it was essentially inactive as an antimitotic agent, only inhibiting tubulin assembly by 40% at 10 μM. In contrast, compound **5** had more potent antitubulin activity (IC_50_ = 9.9 μM) but had less antiproliferative potency (IC_50_ values of 10 and 16 μM against the HeLa and A549 cell lines, respectively).

The same authors designed another series of 2,7-diaryl-[1,2,4]triazolo [1,5-*a*]pyrimidine derivatives as potential tubulin polymerization inhibitors by removing the 5-methyl and 6-acetyl groups of compounds **4** and **5**, with a concomitant switching of substituents at the 2- and 7-positions of the triazolopyrimidine ring [31]. Compound **6**, characterized by the presence of 4′-methoxy-3′-hydroxyphenyl and 3′,4′,5′-trimethoxyphenyl moieties at the 2- and 7-positions of the triazolopyrimidine scaffold, respectively (cf. structure of CA-4), displayed low nanomolar antiproliferative activity (IC_50_, 60 nM) on HeLa cells and activities at micromolar levels against other cells. Compound 6 inhibited tubulin polymerization 3-fold more powerfully than CA-4, and these data proved that the 3′,4′,5′-trimethoxyphenyl ring located at the 7-position of the triazolopyrimidine system contributed to excellent activity.

Taken together, these observations suggest that different triazolopyrimidine congeners may interact with tubulin/microtubules in different manners and with different activities, depending on the particular substitution pattern at the 2-, 5-, 6- and 7-positions, thereby producing different effects on microtubule structure and function.

Thus, in an attempt to identify compounds with improved activity, the structural refinement of compounds **4–6** led us to prepare two novel series of synthetic inhibitors of tubulin polymerization with general structure **7** and **8** (Figure 2), conserving the same [1,2,4]triazolo [1,5-*a*]pyrimidine skeleton known to be linked to microtubule inhibitory activity when the nucleus was modified at its 2- and 7-positions. These compounds were prepared by a facile and efficient three-step procedure, starting from 3-substituted-5-amino-1,2,4-triazoles.

In the first series (compounds **7a–ad**), characterized by a common 7-(3′,4′,5′-trimethoxyanilino)-[1,2,4]triazolo [1,5-*a*]pyrimidine scaffold, the 3′,4′,5′-trimethoxyphenyl ring attached at the 7-position of the triazolopyrimidine nucleus by an anilinic nitrogen (NH), with hydrogen-bond-accepting and donating capability, was conserved. Structure–activity relationships (SARs) were explored by examining various substitutions with electron-withdrawing (F, Cl, CN and CF_3_) or electron-releasing (Me and MeO) groups (EWG’s and ERG’s, respectively) on the phenyl ring attached directly at the 2-position of the triazolopyrimidine scaffold (compounds **7a–c**) or connected by different linkers with length from one to three atoms. These included an amino (NH) group as in anilines **7d–h**, a methyleneamino (CH_2_NH) spacer as in benzylamines **7i** and **7l–y** or its ethyleneamino [CH_2_)_2_NH] homologue as in 2′-phenethylamines **7z–ac**. In addition, the 4′- and 3′-pyridylmethylenamines (compounds **7j** and **7k**, respectively) and 3′-phenylpropylamino (**7ad**) side chains were inserted at the 2-position of the triazolopyrimidine nucleus.

The 3,4,5-trimethoxyphenyl motif is widely accepted to be an important but not essential structural requirement for the optimal biological activity in many colchicine site antimitotic agents [32,33]. Thus, to maintain or further improve the activity of the most potent compounds **7n**, **7p**, **7y** and **7ad** identified in the first series (see Table 1), we explored the possibility of replacing the trimethoxyphenyl pharmacophore of the 3,4,5-trimethoxyanilino unit at the 7-position of the triazolopyrimidine skeleton with substituted anilines, to furnish a second series of 7-anilino triazolopyrimidine derivatives **8a–v**.

## 2. Materials and Methods

### 2.1. Chemistry

^1^H experiments were recorded on either a Bruker AC 200 or a Bruker Avance III 400 spectrometer, while ^13^C NMR spectra were recorded on a Varian 400 Mercury Plus or a Bruker Avance III 400 spectrometer. Chemical shifts (δ) are given in ppm upfield, and the spectra were recorded in appropriate deuterated solvents, as indicated. Mass spectra were recorded by an ESI single quadrupole mass spectrometer (Waters ZQ 2000; Waters Instruments, UK), and the values are expressed as [M + 1]^+^. Melting points (mp) were determined on a Buchi-Tottoli apparatus and are uncorrected. All products reported showed ^1^H and ^13^C NMR spectra in agreement with the assigned structures. The purity of tested compounds was determined by combustion elemental analyses conducted by the Microanalytical Laboratory of the Chemistry Department of the University of Ferrara with a Yanagimoto MT-5 CHN recording elemental analyzer. All tested compounds yielded data consistent with a purity of at least 95% as compared with the theoretical values. Reaction courses and product mixtures were routinely monitored by TLC on silica gel (precoated F254 Merck plates), and compounds were visualized with aqueous KMnO_4_. Flash chromatography was performed using 230–400 mesh silica gel and the indicated solvent system. Organic solutions were dried over anhydrous Na_2_SO_4_.

### 2.2. Biological Assays and Computational Studies

#### 2.2.1. Cell Growth Conditions and Antiproliferative Assay for Human Cancer Cell Lines

Colon adenocarcinoma (HT-29) and T-cell leukemia (Jurkat) cells were grown in RPMI-1640 medium (Gibco, Milano, Italy). Promyelocytic leukemia cells (HL-60), non-small cell lung carcinoma (A549), cervix carcinoma (HeLa) and breast adenocarcinoma (MDA-MB-231) cells were grown in DMEM medium (Gibco, Milano, Italy). Both media were supplemented with 115 units/mL of penicillin G (Gibco, Milano, Italy), 115 μg/mL of streptomycin (Invitrogen, Milano, Italy) and 10% fetal bovine serum (Invitrogen, Milano, Italy). All the cell lines used were purchased from the American Type Culture Collection (ATCC, Manassas, VA, USA). Stock solutions (10 mM) of the compounds were obtained by dissolving them in DMSO.

The cells were seeded in 96-well plates at the appropriate density for each cell line. After 24 h, the cells were treated by performing serial 5-fold dilutions of the tested compounds starting from a concentration of 10 μM. All experimental conditions were tested in triplicate for statistical analysis. The plates were incubated for 72 h at 37 °C in a humidified 5% CO_2_ atmosphere. At the end of the incubation period, 10 μL of 100 μg/mL resazurin solution was added to each well; then, the plate was re-incubated for 3–4 h. The fluorescence of the wells in each plate was monitored using a Spark 10 M spectrophotometer (Tecan Group Ltd., Mannedorf, Switzerland) with a 535 nm excitation wavelength and a 600 nm emission wavelength.

The IC_50_ was defined as the compound concentration required to inhibit cell proliferation by 50%, in comparison with cells treated with the maximum amount of DMSO (0.25%), which was considered 100% viability.

#### 2.2.2. Cytotoxicity in Human PBLs

Human PBLs were obtained from healthy donors by separation on a Lymphoprep^TM ^(Serumwerk Bernburg AG) gradient and were used for the evaluation of the cytotoxic potential of compounds under study in normal human cells. Buffy coats were obtained from the Blood Transfusion Service, Azienda Ospedaliera of Padova and provided at this institution for research purposes without identifier. The samples were not obtained specifically for this study, and for this reason ethical approval was not required. Informed consent was obtained from blood donors according to Italian law no. 219 (21 October 2005). Data were treated by the Blood Transfusion Service according to Italian law on personal data management “Codice in materia di protezione dati personali” (Testo Unico D.L. Giugno 30, 2003 196).

After extensive washing with saline solution (BioConcept, Hank’s Buffered Saline Solution), quiescent PBLs were resuspended (1.0 × 10^6^ cells/mL) in RPMI-1640 medium supplemented with 10% fetal bovine serum. Cytotoxicity evaluations were conducted also in cultures of proliferating PBLs, stimulated with 2.5 mg/mL PHA (Irvine Scientific). Both resting and proliferating lymphocytes were seeded in 96-well plates and maintained overnight at 37 °C in a humidified 5% CO_2_ incubator. Serial (5-fold) dilutions of the tested compounds were added, and viability was determined 72 h later by the resazurin viability assay as described above.

#### 2.2.3. Effects on Tubulin Polymerization and on Colchicine Binding to Tubulin

Bovine brain tubulin was purified as described previously [34]. To evaluate the effects of the compounds on tubulin assembly in vitro [35], varying concentrations were preincubated with 10 μM tubulin in 0.8 M monosodium glutamate (pH 6.6) at 30 °C and the reaction mixtures then cooled to 0 °C. After addition of GTP, the mixtures were transferred to 0 °C cuvettes in Beckman Coulter (Brea, CA, USA) DU-7400/DU-7500 recording spectrophotometers equipped with electronic temperature controllers and warmed to 30 °C, and the assembly of tubulin was observed turbidimetrically. The IC_50_ was defined as the compound concentration that inhibited the extent of assembly by 50% after a 20 min incubation. Inhibition of colchicine binding to tubulin was measured as described before [36], except that the reaction mixtures contained 0.5 μM tubulin and 5 μM each of [^3^H]colchicine and test compound. Only one DEAE-cellulose filter was used per sample, and filtration was by gravity.

#### 2.2.4. Molecular Modeling

All molecular-docking studies were performed on a Viglen Genie Intel^®^ Core^TM^ i7-3770 vPro CPU@ 3.40 GHz × 8 running Ubuntu 18.04. Molecular Operating Environment (MOE) 2020.09 [37] and Maestro (Schrödinger Release 2021-1) [38] were used as molecular modeling software. The tubulin structure was retrieved from the PDB data bank (http://www.rcsb.org/; PDB code 4O2B- accessed on 10 October 2021). The protein was pre-processed using the Schrödinger Protein Preparation Wizard by assigning bond orders, adding hydrogens, and performing a restrained energy minimization of the added hydrogens using the OPLS_2005 force field. Ligand structures were built with MOE and then prepared using the Maestro LigPrep tool by energy minimizing the structures (OPLS_2005 force field), generating possible ionization states at pH 7 ± 2, tautomers and low-energy ring conformers. After isolating a tubulin dimer structure, a 12 Å docking grid (inner-box 10 Å and outer-box 22 Å) was prepared using as centroid the co-crystallized colchicine. Molecular-docking studies were performed using Glide SP Precision keeping the default parameters and setting 10 as number of output poses per input ligand to include in the solution. The output database was saved as a mol2 file. The docking results were visually inspected to evaluate the ability of modeled compounds to bind in the active site.

#### 2.2.5. Cell Cycle Analysis

HeLa cells were seeded at 100,000 cells per well on 6-well tissue culture plates. After 24 h, cells were treated with different test compound concentrations or DMSO for another 24 h. Cells were then fixed and stained with propidium iodide (PI). The fluorescence intensity of 10,000 single-cell events was measured with a Beckman Coulter Cytomics FC500 instrument (Beckman Coulter Italia, Milano, Italy) and the data were analyzed with MultiCycle for Windows software from Phoenix Flow Systems (San Diego, CA, USA).

#### 2.2.6. Measurement of Apoptosis by Flow Cytometry

Surface exposure of phosphatidylserine on apoptotic cells was measured by flow cytometry with a Beckman Coulter Cytomics FC500 (Beckman Coulter Italia, Milano, Italy) by adding annexin-V conjugated to fluorescein isothiocyanate (FITC) to cells according to the manufacturer’s instructions (Annexin-V Fluos, Roche Diagnostic, Monza, Italy). Simultaneously, the cells were stained with PI. Excitation was set at 488 nm, and the emission filters were at 525 and 585 nm, respectively, for FITC and PI.

#### 2.2.7. Measurement of Mitochondrial Membrane Potential and Reactive Oxygen Species (ROS) Production

The mitochondrial potential was evaluated in HeLa cells by flow cytometry using the cell membrane permeable cationic dye 5,5′,6,6′-tetrachloro-1,1′,3,3′-tetraethyl-imidacarbocyanine iodide (JC-1) (Molecular Probes) as described previously [39]. The production of ROS was measured in HeLa cells by flow cytometry using 2′,7′-dichlorodihydrofluorescein diacetate (H_2_DCFDA) (Molecular Probes) as described previously [39].

#### 2.2.8. 5-Ethynyl-2′-deoxyuridine (EdU) Cell Proliferation Assay

HeLa cells were seeded in a 6-well plate and, after a 24 h incubation, were treated with the desired compounds as indicated in the Biological Results and Discussion section. Cells were re-incubated for one day and then labeled with EdU (10 μM) for 3 h under cell culture conditions. Cells were harvested by centrifugation before proceeding with the fixation step using 3.7% formaldehyde. Next, permeabilization with 0.5% Triton X-100 was performed. The reaction cocktail was prepared and added to each well (100 μL/well) following the manufacturer’s instructions (EdU-Click 488, base click, Sigma-Aldrich. Milano Italy), and the plate was incubated at room temperature for 30 min in the dark. Cells were washed 2 times with phosphate-buffered saline (PBS) before reading samples with a Beckman Coulter Cytomics FC500 instrument (λ_ex_/λ_em_ = 496/516 nm).

#### 2.2.9. Evaluation of Cellular Protein Expression with Western Blots

Following growth for various times in the presence of **8q**, HeLa cells were harvested by centrifugation and washed twice in 0 °C PBS. Cells were lysed with 0.1% (*v*/*v*) Triton X-100 containing RNase A at 0 °C, and supernatants were obtained by centrifuging the lysed cells at 15,000× *g* for 10 min at 4 °C. The protein content of the solutions was measured, and 10 μg of protein from each sample was subjected to sodium dodecyl sulfate polyacrylamide gel electrophoresis. Proteins were transferred by electroblotting to a poly(vinylidene difluoride) Hybond-P membrane from GE Healthcare. Membranes were treated with 5% bovine serum albumin in PBS containing 0.1% Tween 20 overnight at 4 °C. The membranes were then exposed for 2 h at room temperature to primary antibodies directed against cyclin B cdc25c, cdc2 (Y15), BCL-2, cleaved poly (ADP-ribose) polymerase (PARP), MCL-1, XIAP and vinculin (all from Cell Signaling), and subsequently for 1 h to peroxidase-labeled secondary antibodies. The membranes were visualized using ECL select (GE Healthcare), and images were acquired using a Uvitec-Alliance imaging system. Densitometric analysis of Western blots was performed by ImageJ software and the results were normalized to β-actin and represented as fold change with respect to untreated controls.

#### 2.2.10. Scratch-Migration Assay

HeLa cells were plated at 100,000 cells per well in a 6-well plate and incubated at 37 °C for 24 h. The day after seeding, cells were nearly confluent, so the sheet of cells was gently wounded through the horizontal and vertical axes using a micropipette tip. Cells were rinsed twice with PBS solution to remove cell debris; next, they were treated with different sublethal concentrations of compound **8q**. At different time points, four images per experimental condition were captured using a stereomicroscope at 10X magnification. The distance between the two edges of the wound area was quantified with ImageJ software.

#### 2.2.11. Immunofluorescence Labeling of Microtubules

HeLa cells were seeded at 15,000 cells/well in a NuncTM Lab-Tek 8-well Chamber SlideTM System (Thermo Fisher Scientific) and incubated under cell culture conditions for 24 h. The cells were treated with various concentrations of compounds or solvent (DMSO as control) and incubated for another 24 h.

After the pharmacological treatment, cells were rinsed with PBS and fixed with 200 μL of 4% paraformaldehyde in PBS for 15 min at room temperature. Cells were then washed twice with PBS 1X before proceeding with permeabilization and protein saturation using a blocking buffer containing 3% bovine serum albumin and 0.1% Triton X-100. Cells were maintained in the blocking buffer for 30 min at room temperature before performing immunostaining of microtubules. Cells were incubated with a murine monoclonal anti-β tubulin antibody (Sigma-Aldrich, Milano, Italy) overnight at 4 °C.

After three washes with PBS 1X, cells were incubated for 1 h at room temperature in the dark with a secondary antibody conjugated to Alexa FluorTM 488 (Invitrogen, Thermo Fisher Scientific, Waltham, MA, USA)).

At the end of the incubation period with the secondary antibody, cells were stained 10 min with 4′,6-diamidino-2-phenylindole (DAPI, 1:10,000, Sigma-Aldrich, Milan, Italy) for counterstaining of the nuclei. Cells were washed three times with PBS and once with MilliQ water before chamber removal. Finally, glycerol 50% was added to each well, and then a coverslip placed over to seal the wells.

Alterations in the microtubule cytoskeleton were documented by fluorescence microscopy using a Zeiss LSM 800 with Airyscan Confocal Spectral microscope (100× magnification) (New York, NY, USA). Images were processed using GIMP software

### 2.3. In Vivo Experiments on Zebrafish Model

#### 2.3.1. Husbandry and Maintenance

Zebrafish (*Danio rerio*) embryos were obtained by pair-wise mating using a fish hatch box. The fertilized eggs were collected in Petri dishes containing embryo medium (5 mM NaCl, 0.17 mM KCl, 0.33 mM CaCl_2_, 0.3 mM MgSO_4_, and 0.1% methylene blue) and raised at 28 °C in an incubator until treatments or transplantation. Normally, the developed embryos were dechorionated using forceps before drug treatment. All procedures were conducted following the recommendations and the guidelines of the Animal Use Ethics Committee concerning the protection of animals used for scientific purposes.

#### 2.3.2. Acute Toxicity Assessment in Zebrafish Embryos

Zebrafish embryos at 72 h post fertilization (hpf) were selected for the acute toxicity assay since, at that time, morphogenesis and function of the primary organ systems are complete. Embryos at 72 hpf were assayed in a 12-well plate, 20 larvae/well for each experimental condition.

The zebrafish embryos were treated as described in the Biological Results and Discussion section using drug concentrations selected on the basis of in vitro experimental data. The drugs were added directly to the fish water from 10 mM stock solutions in DMSO of tested compounds. Embryos treated with the highest dose of DMSO were used as a negative control in all assays to confirm that the vehicle did not cause adverse effects on the zebrafish. CA-4P was employed as the reference compound.

The experiment was carried out at a constant temperature (28.5 °C) in the dark, and at 24 and 48 h time points, zebrafish were subjected to visual observation and image acquisition under a stereomicroscope (Nikon SMZ745T; Nikon, Japan). The number of dead zebrafish embryos for each condition was recorded, and morphological defects were evaluated. The survival rate (%) after treatment was calculated.

#### 2.3.3. Xenograft Model: Injection and Treatment

For the transplantation, zebrafish Tg(fli1: EGFP) embryos expressing enhanced green fluorescent protein (EGFP) specifically in vascular endothelial cells, were kept at 28 °C and manually dechorionated. Before microinjections into zebrafish embryos, HeLa cancer cells were stained with VybrantTM DiI Cell-Labeling Solution (Invitrogen, ThermoFisher, Carlsbad, CA, USA) according to the manufacturer’s protocol, in order to visualize and track them inside the larvae. Stained cells (1.0 × 10^6^ cells) were suspended in 20 μL of PBS 1X obtaining a final concentration of 50 cells/nL; next, the cellular suspension was loaded into borosilicate glass capillary needles (OD/ID: 1.00/0.75 mm, World Precision Instruments, Sarasota, FL, USA). Using a pneumatic picopump, about 150–200 cells were injected within the Duct of Cuvier of 48 hpf embryos anesthetized with tricaine (0.02%, Sigma-Aldrich, Milano, Italy) and mounted in low-melting-point agarose (1%).

After injection, xenograft-harboring larvae were incubated to recover at 34 °C in darkness, in fish water containing 1-phenyl 2-thiourea (PTU) at the final concentration of 0.003% *w*/*v* to inhibit the pigmentation process. At 2 h post-injection, embryos were examined to assure the homogeneity of the xenografts. Only successfully injected xenografted larvae, with approximately 100 HeLa red-stained cells spread in the caudal area, were selected for the drug treatments. After selection of the cohort, embryos were distributed to 96-well plates with one embryo placed in each well. The injected xenografts were exposed to the doses of **8q** indicated in the Biological Results and Discussion section and maintained at 34 °C. Considering that the zebrafish is a poikilothermic fish that preferably grows around 28 °C while the optimal temperature for human cell growth is 37 °C, after the transplantation the xenografted animals were maintained at 34 °C as a compromise (and a tolerated temperature for zebrafish) [40].

DMSO was used as a vehicle control. For drug treatments, **8q** was dissolved in DMSO and diluted directly into embryo medium.

Initially (time 0 h pre-treatment) and after one day post-treatment, the tumors were photographed with a Zeiss AxioObserver microscope for live-cell imaging. HeLa cell fluorescence was recorded to investigate reduction, death and migration of tumor cells. Using Fiji ImageJ software, the reduction in cancer cell fluorescence, related to the effectiveness of the anticancer drug treatments, was calculated and plotted. The fluorescence of at least 20 identically treated embryos was compared to vehicle control set to 100%.

#### 2.3.4. Statistical Analysis

Graphs and statistical analyses were performed using GraphPad Prism software (v. 7.0, GraphPad, La Jolla, CA, USA). All data in graphs represented the mean of at least three independent experiments ± SEM. Statistical significance was determined using Student’s *t*-test or ANOVA (one- or two-way) depending on the type of data. Asterisks indicate a significant difference between the treated and the control group, unless otherwise specified. * *p* < 0.05, ** *p* < 0.01, *** *p* < 0.001, **** *p* < 0.0001.

## 3. Results and Discussion

### 3.1. Chemistry

#### 3.1.1. Synthetic Approach for the Preparation of 2,7-Disubstituted [1,2,4]triazolo [1,5-a]pyrimidines **7a–ad** and **8a–v**

Our new target compounds **7a–ad**, based on 3′,4′,5′-trimethoxyaniline moiety at the 7-position of [1,2,4]triazolo [1,5-a]pyrimidine scaffold, were prepared using a three-step synthetic approach as shown in Figure 1. The cyclization reaction of 3-substituted-5-amino-1,2,4-triazoles **9a–ad** with ethyl acetoacetate in acetic acid as solvent under reflux afforded the corresponding 2-substituted-7-hydroxy-[1,2,4]triazolo [1,5-*a*]pyrimidines **10a–ad** in 70–95% yield according to a literature method [41]. The subsequent chlorination reaction of **10a–ad** with a mixture of phosphorus oxychloride and pyridine at 60 °C provided the chlorinated analogues **11a–ad**, which were reacted for nucleophilic substitution reactions with 3,4,5-trimethoxyaniline in refluxing isopropanol to furnish final compounds **7a–ad**.

Similarly, final compounds **8a–v** characterized by the presence of substituted anilines at the 7-position of [1,2,4]triazolo [1,5-*a*]pyrimidine scaffold, were achieved by the condensation of intermediates **11n**, **11p**, **11y** and **11ad** with different commercially available anilines in refluxing isopropanol as shown in Figure 2.

3-Substituted-5-amino-1,2,4-triazole intermediates **9a–ad** used for preparation of final products were synthesized according to synthetic procedures previously described in the literature, and their preparation and characterization are described in the Appendix A. Synthetic, physical and analytical data of new compounds, which have not been previously published (**9j–m**, **9o–r**, **9t–u**, **9w–y** and **9aa–ac**), are presented in the Appendix A.

#### 3.1.2. General Procedure A for the Synthesis of Compounds **10a–ad**

A mixture of the appropriate 3-substituted-1*H*-1,2,4-triazole-5-amine **9a–ad** (5 mmol) and 3-oxobutanoic acid ethyl ester (1.27 mL, 10 mmol, 2 equiv.) in acetic acid (25 mL) was heated at 80 °C for 18 h. Volatiles were removed, the crude residue suspended in EtOAc was stirred for 15 min and then filtered, giving the appropriate 2-substituted-5-methyl-[1,2,4]triazolo [1,5-*a*]pyrimidin-7-ol **10a–ad** as the solid recovered by filtration. Chemical, physical and analytical characterization of the synthesized compounds **10a–ad** are reported in the Appendix A.

#### 3.1.3. General Procedure B for the Preparation of Compounds **11a–ad**

A mixture of the appropriate 2-substituted-5-methyl-[1,2,4]triazolo [1,5-*a*]pyrimidin-7-ol **10a–ad** (2.2 mmol), pyridine (89 uL, 1.08 mmol) and phosphorus(V) oxychloride (2.0 mL, 2.14 mmol) was stirred at 60 °C for 6 h; then, volatiles were removed. Dichloromethane was added, followed by a 10% aqueous solution of sodium carbonate. The phases were separated, and the organic phase was washed with brine, dried over sodium sulfate, filtered and concentrated *in vacuo*. The crude products were stirred for 15 min with ethyl ether, and the mixtures were filtered to furnish the desired 2-substituted-5-methyl-7-chloro-[1,2,4]triazolo [1,5-*a*]pyrimidine derivatives **11a–d**. Chemical, physical and analytical characterization of the synthesized compounds **11a–ad** are reported in the Appendix A.

#### 3.1.4. General Procedure C for the Synthesis of Compounds **7a–ad** and **8a–v**

A mixture of the appropriate 2-substituted-5-methyl-7-chloro-[1,2,4]triazolo [1,5-*a*]pyrimidine **11a–ad** (0.780 mmol) and the appropriate substituted aniline (1.58 mmol, 2 equiv.) in *iso*-propanol (15 mL) was stirred at 80 °C for 3 h and then evaporated to dryness *in vacuo*. The residue was portioned between dichloromethane and water, and the organic phase was separated, washed with brine, dried over sodium sulfate, filtered and concentrated. The crude residue was purified by flash chromatography on silica gel to furnish the desired derivative.

2-(4-Fluorophenyl)-5-methyl-N-(3,4,5-trimethoxyphenyl)-[1,2,4]triazolo [1,5-*a*]pyrimidin-7-amine (**7a**)

Following general procedure C, using 3′,4′,5′-trimethoxyaniline as the aniline, the crude residue was purified by flash chromatography, using ethyl acetate/petroleum ether 8:2 as eluent, to furnish **7a** as a light purple solid. Yield: 56%, mp 253 °C. ^1^H-NMR (DMSO-*d_6_*) δ: 2.44 (s, 3H), 3.70 (s, 3H), 3.80 (s, 6H), 6.46 (s, 1H), 6.81 (s, 2H), 7.39 (t, *J* = 8.8 Hz, 2H), 8.27–8.31 (m, 2H), 9.97 (s, 1H). ^13^C-NMR (DMSO-*d_6_*) δ: 24.71, 55.92 (2C), 59.99, 89.73, 102.45 (2C), 115.66 and 115.87 (*J*_2CF_ = 21.3 Hz, 2C), 127.31, 128.93 and 129.01 (*J*_3CF_ = 8.3 Hz, 2C), 132.34, 135.52, 145.73, 153.17 (2C), 156.14, 162.06 and 164.41 (*J*_1CF_ = 245 Hz, 1C), 162.12, 164.27. MS (ESI): [M+1]^+^ = 410.2. Anal. calcd for C_21_H_20_FN_5_O_3_. C, 61.61; H, 4.92; N, 11.11; found: C, 61.47; H, 4.74; N, 10.97.

2-(4-Chlorophenyl)-5-methyl-N-(3,4,5-trimethoxyphenyl)-[1,2,4]triazolo [1,5-a]pyrimidin-7-amine (**7b**)

Following general procedure C, using 3′,4′,5′-trimethoxyaniline as the aniline, the crude residue was purified by flash chromatography, using ethyl acetate/petroleum ether 7:3 as eluent, to furnish **7b** as a gray solid. Yield: 63%, mp 260 °C. ^1^H-NMR (DMSO-*d_6_*) δ: 2.44 (s, 3H), 3.70 (s, 3H), 3.80 (s, 6H), 6.47 (s, 1H), 6.81 (s, 2H), 7.63 (d, *J* = 8.4 Hz, 2H), 8.23 (dd, *J* = 8.4 Hz, 2H), 9.98 (s, 1H). ^13^C-NMR (DMSO-*d_6_*) δ: 24.72, 55.93 (2C), 60.00, 89.92, 102.46 (2C), 128.42 (2C), 128.88 (2C), 129.65, 132.32, 134.83, 135.55, 145.76, 153.17 (2C), 156.14, 161.97, 164.41. MS (ESI): [M+1]^+^ = 426.5. Anal. calcd for C_21_H_20_ClN_5_O_3_. C, 59.23; H, 4.73; N, 16.44; found: C, 59.01; H, 4.59; N, 16.33.

5-Methyl-2-(p-tolyl)-N-(3,4,5-trimethoxyphenyl)-[1,2,4]triazolo [1,5-a]pyrimidin-7-amine (**7c**)

Following general procedure C, using 3′,4′,5′-trimethoxyaniline as the aniline, the crude residue was purified by flash chromatography, using ethyl acetate/petroleum ether 7:3 as eluent, to furnish **7c** as a purple solid. Yield: 80%, mp 248 °C. ^1^H-NMR (DMSO-*d_6_*) δ: 2.39 (s, 3H), 2.44 (s, 3H), 3.70 (s, 3H), 3.80 (s, 6H), 6.45 (s, 1H), 6.82 (s, 2H), 7.36 (d, *J* = 8.0 Hz, 2H), 8.14 (d, *J* = 8.0 Hz, 2H), 9.96 (bs, 1H). ^13^C-NMR (DMSO-*d_6_*) δ: 20.95, 24.70, 55.94 (2C), 60.00, 89.58, 102.42 (2C), 126.71 (2C), 128.02, 129.26 (2C), 132.40, 135.49, 139.82, 145.66, 153.17 (2C), 156.07, 163.05, 164.06. MS (ESI): [M+1]^+^ = 406.54. Anal. calcd for C_22_H_23_N_5_O_3_. C, 65.17; H, 5.72; N, 17.27; found: C, 64.97; H, 5.61; N, 17.11.

5-Methyl-N2-phenyl-N7-(3,4,5-trimethoxyphenyl)-[1,2,4]triazolo [1,5-a]pyrimidine-2,7-diamine (**7d**)

Following general procedure C, using 3′,4′,5′-trimethoxyaniline as the aniline, the crude residue was purified by flash chromatography, using ethyl acetate/petroleum ether 6:4 as eluent, to furnish **7d** as a beige solid. Yield: 56%, mp 241 °C. ^1^H-NMR (DMSO-*d_6_*) δ: 2.35 (s, 3H), 3.64 (s, 3H), 3.79 (s, 6H), 6.24 (s, 1H), 6.78 (s, 2H), 6.87 (t, *J* = 8.0 Hz, 1H), 7.26 (t, *J* = 8.0 Hz, 2H), 7.81 (d, *J* = 8.0 Hz, 2H), 9.53 (s, 1H), 9.59 (s, 1H). ^13^C-NMR (DMSO-*d_6_*) δ: 24.45, 55.94 (2C), 60.02, 88.80, 102.95 (2C), 116.73 (2C), 119.86, 128.52 (2C), 132.41, 135.60, 140.97, 144.96, 153.14 (2C), 154.29, 161.90, 162.64. MS (ESI): [M+1]^+^ = 407.4. Anal. calcd for C_21_H_22_N_6_O_3_. C, 62.06; H, 5.46; N, 20.68; found: C, 61.88; H, 5.33; N, 20.59.

N2-(4-Fluorophenyl)-5-methyl-N7-(3,4,5-trimethoxyphenyl)-[1,2,4]triazolo [1,5-a]pyrimidine-2,7-diamine (**7e**)

Following general procedure C, using 3′,4′,5′-trimethoxyaniline as the aniline, the crude residue was purified by flash chromatography, using ethyl acetate/petroleum ether 9:1 as eluent, to furnish **7e** as a pink solid. Yield: 53%, mp 247 °C. ^1^H-NMR (DMSO-*d_6_*) δ: 2.36 (s, 3H), 3.71 (s, 3H), 3.81 (s, 6H), 6.25 (s, 1H), 6.80 (s, 2H), 7.09 (t, *J* = 8.8 Hz, 2H), 7.85–7.87 (m, 2H), 9.58 (s, 1H), 9.65 (s, 1H).^13^C-NMR (DMSO-*d_6_*) δ: 24.45, 55.95 (2C), 60.03, 88.82, 102.98 (2C), 114.81 and 115.03 (*J*_2CF_ = 22.1 Hz, 2C), 118.00 and 118.07 (*J*_3CF_ = 6.9 Hz, 2C), 132.37, 135.65, 137.48, 144.98, 153.17 (2C), 154.30, 155.03 and 157.38 (*J*_1CF_ = 223 Hz, 1C), 161.83, 162.70. MS (ESI): [M+1]^+^ = 425.3. Anal. calcd for C_21_H_21_FN_6_O_3_. C, 59.43; H, 4.99; N, 19.80; found: C, 59.21; H, 4.86; N, 19.70.

N2-(4-Chlorophenyl)-5-methyl-N7-(3,4,5-trimethoxyphenyl)-[1,2,4]triazolo [1,5-a]pyrimidine-2,7-diamine (**7f**)

Following general procedure C, using 3′,4′,5′-trimethoxyaniline as the aniline, the crude residue was purified by flash chromatography, using ethyl acetate/petroleum ether 8:2 as eluent, to furnish **7f** as a pink solid. Yield: 56%, mp 251 °C. ^1^H-NMR (DMSO-*d_6_*) δ: 2.37 (s, 3H), 3.71 (s, 3H), 3.81 (s, 6H), 6.26 (s, 1H), 6.80 (s, 2H), 7.28 (d, *J* = 8.8 Hz, 2H), 7.86 (d, *J* = 8.8 Hz, 2H), 9.61 (s, 1H), 9.81 (s, 1H). ^13^C-NMR (DMSO-*d_6_*) δ: 24.49, 55.98 (2C), 60.05, 88.95, 103.06 (2C), 118.32 (2C), 123.36, 128.30 (2C), 132.34, 135.71, 139.98, 145.08, 153.20 (2C), 154.32, 161.63, 162.86. MS (ESI): [M+1]^+^ = 441.2. Anal. calcd for C_21_H_21_ClN_6_O_3_. C, 57.21; H, 4.80; N, 19.06; found: C, 57.02; H, 4.70; N, 18.98.

5-Methyl-N2-(p-tolyl)-N7-(3,4,5-trimethoxyphenyl)-[1,2,4]triazolo [1,5-a]pyrimidine-2,7-diamine (**7g**)

Following general procedure C, using 3′,4′,5′-trimethoxyaniline as the aniline, the crude residue was purified by flash chromatography, using ethyl acetate/petroleum ether 8:2 as eluent, to furnish **7g** as a purple solid. Yield: 60%, mp 234 °C. ^1^H-NMR (DMSO-*d_6_*) δ: 2.23 (s, 3H), 2.34 (s, 3H), 3.69 (s, 3H), 3.79 (s, 6H), 6.22 (s, 1H), 6.78 (s, 2H), 7.05 (d, *J* = 8.4 Hz, 2H), 7.69 (d, *J* = 8.4 Hz, 2H), 9.45 (s, 1H), 9.51 (s, 1H). ^13^C-NMR (DMSO-*d_6_*) δ: 20.89, 25.08, 56.58 (2C), 60.67, 89.38, 103.57 (2C), 117.40 (2C), 129.07, 129.68 (2C), 133.08, 136.22, 139.17, 145.55, 153.78 (2C), 154.97, 162.66, 163.17. MS (ESI): [M+1]^+^ = 421.4. Anal. calcd for C_22_H_24_N_6_O_3_. C, 62.84; H, 5.75; N, 19.99; found: C, 62.71; H, 5.55; N, 19.78.

N2-(4-Methoxyphenyl)-5-methyl-N7-(3,4,5-trimethoxyphenyl)-[1,2,4]triazolo [1,5-a]pyrimidine-2,7-diamine (**7h**)

Following general procedure C, using 3′,4′,5′-trimethoxyaniline as the aniline, the crude residue was purified by flash chromatography, using ethyl acetate/petroleum ether 9:1 as eluent, to furnish **7h** as a gray solid. Yield: 48%, mp 215 °C. ^1^H-NMR (DMSO-*d_6_*) δ: 2.36 (s, 3H), 3.70 (s, 3H), 3.71 (s, 3H), 3.80 (s, 6H), 6.24 (s, 1H), 6.79 (s, 2H), 6.84 (d, *J* = 6.2 Hz, 2H), 7.74 (d, *J* = 6.2 Hz, 2H), 9.39 (s, 1H), 9.51 (s, 1H). ^13^C-NMR (DMSO-*d_6_*) δ: 24.46, 55.03, 55.94 (2C), 60.04, 88.69, 102.86 (2C), 113.80 (2C), 117.99 (2C), 132.46, 134.51, 136.00, 144.82, 153.01, 153.14 (2C), 154.36, 162.12, 162.46. MS (ESI): [M+1]^+^ = 437.4. Anal. calcd for C_22_H_24_N_6_O_4_. C, 60.54; H, 5.54; N, 19.25; found: C, 60.41; H, 5.38; N, 19.14.

N^2^-Benzyl-5-methyl-N^7^-(3,4,5-trimethoxyphenyl)-[1,2,4]triazolo [1,5-*a*]pyrimidine-2,7-diamine (**7i**)

Following general procedure C, using 3′,4′,5′-trimethoxyaniline as the aniline, the crude residue was purified by flash chromatography, using ethyl acetate:methanol 9.5:0.5 as eluent, to furnish **7i** as a cream-colored solid. Yield: 62%, mp 173–175 °C. ^1^H-NMR (DMSO-*d_6_*) δ: 2.30 (s, 3H), 3.66 (s, 3H), 3.76 (s, 6H), 4.46 (d, *J* = 6.4 Hz, 2H), 6.21 (s, 1H), 6.72 (s, 2H), 7.13 (t, *J* = 6.4 Hz, 1H), 7.20 (t, *J* = 8.4 Hz, 1H), 7.31 (t, *J* = 8.4 Hz, 2H), 7.40 (d, *J* = 8.4 Hz, 2H), 9.24 (bs, 1H). ^13^C-NMR (DMSO-*d_6_*) δ: 24.95, 46.00, 56.45 (2C), 60.54, 89.03, 102.83 (2C), 126.99, 127.60 (2C), 128.59 (2C), 133.24, 135.78, 141.15, 144.79, 153.62 (2C), 155.92, 162.09, 166.55. MS (ESI): [M+1]^+^ = 421.3. Anal. calcd for C_22_H_24_N_6_O_3_. C, 62.84; H, 5.75; N, 19.99; found: C, 62.68; H, 5.54; N, 19.71.

5-Methyl-N^2^-(pyridin-4-ylmethyl)-N^7^-(3,4,5-trimethoxyphenyl)-[1,2,4]triazolo [1,5-*a*]pyrimidine-2,7-diamine (**7j**)

Following general procedure C, using 3′,4′,5′-trimethoxyaniline as the aniline, the crude residue was purified by flash chromatography, using ethyl acetate:methanol 9:1 as eluent, to furnish **7j** as a brownish foam. Yield: 38%, mp 184–186 °C. ^1^H-NMR (DMSO-*d_6_*) δ: 2.31 (s, 3H), 3.68 (s, 3H), 3.78 (s, 6H), 4.55 (d, *J* = 6.4 Hz, 2H), 6.22 (s, 1H), 6.72 (s, 2H), 7.23 (t, *J* = 6.4 Hz, 1H), 7.37 (d, *J* = 4.4 Hz, 2H), 8.46 (dd, *J* = 4.4 and 1.6 Hz, 2H), 9.26 (s, 1H). ^13^C-NMR (DMSO-*d_6_*) δ: 24.95, 45.07, 56.44 (2C), 60.54, 89.15, 102.87 (2C), 122.60 (2C), 133.19, 135.81, 144.88, 149.86 (2C), 150.22, 153.62 (2C), 155.92, 162.24, 166.34. MS (ESI): [M+1]^+^ = 422.3. Anal. calcd for C_21_H_23_N_7_O_3_. C, 59.85; H, 5.50; N, 23.26; found: C, 59.69; H, 5.38; N, 23.16.

5-Methyl-N^2^-(pyridin-3-ylmethyl)-N^7^-(3,4,5-trimethoxyphenyl)-[1,2,4]triazolo [1,5-*a*]pyrimidine-2,7-diamine (**7k**)

Following general procedure C, using 3′,4′,5′-trimethoxyaniline as the aniline, the crude residue was purified by flash chromatography, using ethyl acetate:methanol 9:1 as eluent, to furnish **7k** as a brownish foam. Yield: 34%, mp 178–180 °C. ^1^H-NMR (DMSO-*d_6_*) δ: 2.31 (s, 3H), 3.69 (s, 3H), 3.78 (s, 6H), 4.52 (d, *J* = 6.2 Hz, 2H), 6.20 (s, 1H), 6.72 (s, 2H), 7.18 (s, 1H), 7.35 (dd, *J* = 7.6 and 4.6 Hz, 1H), 7.81 (d, J = 7.6 Hz, 1H), 8.44 (d, *J* = 3.2 Hz, 1H), 8.64 (s, 1H), 9.26 (s, 1H). MS (ESI): [M+1]^+^ = 422.3. Anal. calcd for C_21_H_23_N_7_O_3_. C, 59.85; H, 5.50; N, 23.26; found: C, 59.71; H, 5.33; N, 23.16.

N^2^-(4-Fluorobenzyl)-5-methyl-N^7^-(3,4,5-trimethoxyphenyl)-[1,2,4]triazolo [1,5-*a*]pyrimidine-2,7-diamine (**7l**)

Following general procedure C, using 3′,4′,5′-trimethoxyaniline as the aniline, the crude residue was purified by flash chromatography, using ethyl acetate:methanol 9.5:0.5 as eluent, to furnish **7l** as a brownish foam. Yield: 59%, mp 156–158 °C. ^1^H-NMR (DMSO-*d_6_*) δ: 2.32 (s, 3H), 3.69 (s, 3H), 3.78 (s, 6H), 4.50 (d, *J* = 6.4 Hz, 2H), 6.22 (s, 1H), 6.73 (s, 2H), 7.06–7.19 (m, 3H), 7.40–7.49 (m, 2H), 9.25 (s, 1H). ^13^C-NMR (DMSO-*d_6_*) δ: 24.37, 44.74, 55.88 (2C), 59.97, 88.47, 102.28 (2C), 114.60 and 114.80 (*J*_2CF_ = 21 Hz, 2C), 128.98 and 129.05 (*J*_3CF_ = 7.6 Hz, 2C), 132.63, 135.32, 136.74, 144.23, 153.05 (2C), 155.33, 161.57, 162.17, 165.84. MS (ESI): [M+1]^+^ = 439.2. Anal. calcd for C_22_H_23_FN_6_O_3_. C, 60.27; H, 5.29; N, 19.17; found: C, 60.16; H, 5.08; N, 19.01.

N^2^-(3-Fluorobenzyl)-5-methyl-N^7^-(3,4,5-trimethoxyphenyl)-[1,2,4]triazolo [1,5-*a*]pyrimidine-2,7-diamine (**7m**)

Following general procedure C, using 3′,4′,5′-trimethoxyaniline as the aniline, the crude residue was purified by flash chromatography, using ethyl acetate:methanol 9.5:0.5 as eluent, to furnish **7m** (as a whitish foam. Yield: 45%, mp 176–178 °C. ^1^H-NMR (DMSO-*d_6_*) δ: 2.31 (s, 3H), 3.69 (s, 3H) 3.78 (s, 6H) 4.53 (d, *J* = 6.2 Hz, 2H) 6.21 (s, 1H), 6.72 (s, 2H) 7.05 (t, *J* = 9.8 Hz, 1H) 7.15–7.21 (m, 3H), 7.34–7.40 (m, 1H), 9.25 (s, 1H). ^13^C-NMR (DMSO-*d_6_*) δ: 24.36, 44.97, 55.87 (2C), 59.97, 88.51, 102.27 (2C), 113.04, 113.25, 113.53, 113.74, 123.02, 129.91, 135.19, 143.75, 144.32, 153.05 (2C), 155.36, 161.54, 165.78. MS (ESI): [M+1]^+^ = 439.1. Anal. calcd for C_22_H_23_FN_6_O_3_. C, 60.27; H, 5.29; N, 19.17; found: C, 60.08; H, 5.12; N, 19.02.

N^2^-(4-Chlorobenzyl)-5-methyl-N^7^-(3,4,5-trimethoxyphenyl)-[1,2,4]triazolo [1,5-*a*]pyrimidine-2,7-diamine (**7n**)

Following general procedure C, using 3′,4′,5′-trimethoxyaniline as the aniline, the crude residue was purified by flash chromatography, using ethyl acetate:methanol 9.5:0.5 as eluent, to furnish **7n** as a white solid. Yield: 68%, mp 140–142 °C. ^1^H-NMR (DMSO-*d_6_*) δ: 2.30 (s, 3H), 3.66 (s, 3H), 3.76 (s, 6H), 4.47 (d, *J* = 6.4 Hz, 2H), 6.21 (s, 1H), 6.71 (s, 2H), 7.15 (t, *J* = 6.4 Hz, 1H), 7.34 (d, *J* = 8.4 Hz, 2H), 7.39 (d, *J* = 8.4 Hz, 2H), 9.26 (s, 1H). ^13^C-NMR (DMSO-*d_6_*) δ: 24.95, 45.36, 56.45 (2C), 60.54, 89.06, 102.87 (2C), 128.53 (2C), 129.52 (2C), 131.52, 133.19, 135.82, 140.23, 144.82, 153.62 (2C), 155.90, 162.16, 166.39. MS (ESI): [M+1]^+^ = 455.4. Anal. calcd for C_22_H_23_ClN_6_O_3_. C, 58.09; H, 5.10; N, 18.47; found: C, 57.89; H, 4.98; N, 18.33.

N^2^-(3-Chlorobenzyl)-5-methyl-N^7^-(3,4,5-trimethoxyphenyl)-[1,2,4]triazolo [1,5-*a*]pyrimidine-2,7-diamine (**7o**)

Following general procedure C, using 3′,4′,5′-trimethoxyaniline as the aniline, the crude residue was purified by flash chromatography, using ethyl acetate:methanol 9.5:0.5 as eluent, to furnish **7o** as a yellowish foam. Yield: 64%, mp 136–138 °C. ^1^H-NMR (DMSO-*d_6_*) δ: 2.31 (s, 3H), 3.69 (s, 3H), 3.78 (s, 6H), 4.54 (d, *J* = 6.4 Hz, 2H), 6.22 (s, 1H), 6.72 (s, 2H), 7.17 (t, *J* = 6.4 Hz, 1H), 7.27–7.29 (m, 1H), 7.33–7.34 (m, 2H), 7.41 (s, 1H), 9.27 (s, 1H). ^13^C-NMR (DMSO-*d_6_*) δ: 24.37, 44.82, 55.87 (2C), 59.97, 88.54, 102.29 (2C), 125.74, 126.38, 126.75 (2C), 129.95, 132.73, 135.23, 143.35, 144.31, 153.06 (2C), 155.35, 161.60, 165.75. MS (ESI): [M+1]^+^ = 455.4. Anal. calcd for C_22_H_23_ClN_6_O_3_. C, 58.09; H, 5.10; N, 18.47; found: C, 57.91; H, 4.89; N, 18.25.

5-Methyl-N2-(4-methylbenzyl)-N7-(3,4,5-trimethoxyphenyl)-[1,2,4]triazolo [1,5-*a*]pyrimidine-2,7-diamine (**7p**)

Following general procedure C, using 3′,4′,5′-trimethoxyaniline as the aniline, the crude residue was purified by flash chromatography, using ethyl acetate:methanol 9.5:0.5 as eluent, to furnish **7p** as a white solid. Yield: 63%, mp 146–148 °C. ^1^H-NMR (DMSO-*d_6_*) δ: 2.24 (s, 3H), 2.29 (s, 3H), 3.66 (s, 3H), 3.75 (s, 6H), 4.44 (d, *J* = 6.4 Hz, 2H), 6.20 (s, 1H), 6.71 (s, 2H), 7.03 (t, *J* = 6.4 Hz, 1H), 7.08 (d, *J* = 8.0 Hz, 2H), 7.24 (d, *J* = 8.0 Hz, 2H), 9.23 (s, 1H). ^13^C-NMR (DMSO-*d_6_*) δ: 21.12, 24.95, 45.77, 56.44 (2C), 60.54, 88.99, 102.81 (2C), 127.62 (2C), 129.12 (2C), 133.27, 135.96, 138.07, 140.00, 144.77, 153.62 (2C), 155.90, 162.06, 166.54. MS (ESI): [M+1]^+^ = 435.5. Anal. calcd for C_23_H_26_N_6_O_3_. C, 63.58; H, 6.03; N, 19.34; found: C, 63.44; H, 5.89; N, 19.21.

5-Methyl-N^2^-(3-methylbenzyl)-N^7^-(3,4,5-trimethoxyphenyl)-[1,2,4]triazolo [1,5-*a*]pyrimidine-2,7-diamine (**7q**)

Following general procedure C, using 3′,4′,5′-trimethoxyaniline as the aniline, the crude residue was purified by flash chromatography, using ethyl acetate:methanol 9.75:0.25 as eluent, to furnish **7q** as a white solid. Yield: 64%, mp 162–164 °C. ^1^H-NMR (DMSO-*d_6_*) δ: 2.26 (s, 3H), 2.29 (s, 3H), 3.66 (s, 3H), 3.75 (s, 6H), 4.47 (d, *J* = 6.4 Hz, 2H), 6.21 (s, 1H), 6.71 (s, 2H), 6.96–7.01 (m, 3H), 7.14–7.16 (m, 2H), 9.25 (s, 1H). ^13^C-NMR (DMSO-*d_6_*) δ: 21.54, 24.95, 45.95, 56.44 (2C), 60.54, 89.02, 102.82 (2C), 124.67, 127.62, 128.13, 128.51, 133.23, 135.78, 137.57, 141.08, 144.77, 153.62 (2C), 155.90, 162.08, 162.55. MS (ESI): [M+1]^+^ = 435.2. Anal. calcd for C_23_H_26_N_6_O_3_. C, 63.58; H, 6.03; N, 19.34; found: C, 63.37; H, 5.91; N, 19.20.

5-Methyl-N^2^-(2-methylbenzyl)-N^7^-(3,4,5-trimethoxyphenyl)-[1,2,4]triazolo [1,5-*a*]pyrimidine-2,7-diamine (**7r**)

Following general procedure C, using 3′,4′,5′-trimethoxyaniline as the aniline, the crude residue was purified by flash chromatography, using ethyl acetate:methanol 9.75:0.25 as eluent, to furnish **7r** as a white solid. Yield: 63%, mp 182–184 °C. ^1^H-NMR (DMSO-*d_6_*) δ: 2.29 (s, 3H), 2.31 (s, 3H), 3.66 (s, 3H), 3.75 (s, 6H), 4.47 (d, *J* = 6.4 Hz, 2H), 6.20 (s, 1H), 6.71 (s, 2H), 7.01 (t, *J* = 5.6 Hz, 1H), 7.11–7.13 (m, 3H), 7.38 (t, *J* = 5.6 Hz, 1H), 9.22 (s, 1H). ^13^C-NMR (DMSO-*d_6_*) δ: 19.12, 24.95, 43.98, 56.45 (2C), 60.54, 89.00, 102.89 (2C), 126.08 (2C), 126.92, 127.37, 130.20, 133.21, 135.81, 138.71, 144.82, 153.62 (2C), 155.94, 162.09, 166.57. MS (ESI): [M+1]^+^ = 434.8. Anal. calcd for C_23_H_26_N_6_O_3_. C, 63.58; H, 6.03; N, 19.34; found: C, 63.31; H, 5.88; N, 19.18.

N^2^-(4-Methoxybenzyl)-5-methyl-N^7^-(3,4,5-trimethoxyphenyl)-[1,2,4]triazolo [1,5-*a*]pyrimidine-2,7-diamine (**7s**)

Following general procedure C, using 3′,4′,5′-trimethoxyaniline as the aniline, the crude residue was purified by flash chromatography, using ethyl acetate:methanol 9.5:0.5 as eluent, to furnish **7s** as a white solid. Yield: 63%, mp 182–184 °C. ^1^H-NMR (DMSO-*d_6_*) δ: 2.29 (s, 3H), 3.66 (s, 3H), 3.69 (s, 3H), 3.76 (s, 6H), 4.41 (d, *J* = 6.4 Hz, 2H), 6.21 (s, 1H), 6.71 (s, 2H), 6.84 (d, *J* = 8.4 Hz, 2H), 7.00 (t, *J* = 6.4 Hz, 1H), 7.28 (d, *J* = 8.4 Hz, 2H), 9.25 (bs, 1H). ^13^C-NMR (DMSO-*d_6_*) δ: 24.95, 45.48, 55.46, 56.45 (2C), 60.54, 88.98, 102.82 (2C), 113.98 (2C), 128.96 (2C), 133.04, 133.24, 135.78, 144.75, 153.62 (2C), 155.90, 158.53, 162.06, 166.51. MS (ESI): [M+1]^+^ = 451.4. Anal. calcd for C_23_H_26_N_6_O_4_. C, 61.32; H, 5.82; N, 18.66; found: C, 61.17; H, 5.65; N, 18.44.

N^2^-(3-Methoxybenzyl)-5-methyl-N^7^-(3,4,5-trimethoxyphenyl)-[1,2,4]triazolo [1,5-*a*]pyrimidine-2,7-diamine (**7t**)

Following general procedure C, using 3′,4′,5′-trimethoxyaniline as the aniline, the crude residue was purified by flash chromatography, using ethyl acetate:methanol 9.5:0.5 as eluent, to furnish **7t** as a white solid. Yield: 68%, mp 150–152 °C. ^1^H-NMR (DMSO-*d_6_*) δ: 2.29 (s, 3H), 3.66 (s, 3H), 3.70 (s, 3H), 3.75 (s, 6H), 4.47 (d, *J* = 6.0 Hz, 2H), 6.21 (s, 1H), 6.71 (s, 2H), 6.78 (dd, *J* = 8.4 and 2.0 Hz, 1H), 6.97–7.00 (s, 2H), 7.12 (t, *J* = 8.4 Hz, 1H), 7.20 (t, *J* = 8.4 Hz, 1H), 9.26 (bs, 1H). ^13^C-NMR (DMSO-*d_6_*) δ: 24.95, 45.96, 55.39, 56.45 (2C), 60.54, 89.03, 102.83 (2C), 112.25, 113.34, 119.79, 129.65, 133.22, 135.79, 142.80, 144.79, 153.62 (2C), 155.91, 159.69, 162.11, 166.53. MS (ESI): [M+1]^+^ = 450.7. Anal. calcd for C_23_H_26_N_6_O_4_. C, 61.32; H, 5.82; N, 18.66; found: C, 61.20; H, 5.51; N, 18.41.

N^2^-(2-Methoxybenzyl)-5-methyl-N^7^-(3,4,5-trimethoxyphenyl)-[1,2,4]triazolo [1,5-*a*]pyrimidine-2,7-diamine (**7u**)

Following general procedure C, using 3′,4′,5′-trimethoxyaniline as the aniline, the crude residue was purified by flash chromatography, using ethyl acetate:methanol 9.5:0.5 as eluent, to furnish **7u** as a white solid. Yield: 83%, mp 208–210 °C. ^1^H-NMR (DMSO-*d_6_*) δ: 2.29 (s, 3H), 3.66 (s, 3H), 3.75 (s, 6H), 3.81 (s, 3H), 4.48 (d, J = 6.4 Hz, 2H), 6.20 (s, 1H), 6.70 (s, 2H), 6.84–6.88 (m, 2H), 6.95 (d, *J* = 7.2 Hz, 1H), 7.20 (t, *J* = 7.2 Hz, 1H), 7.27 (dd, *J* = 7.2 and 1.2 Hz, 1H), 9.22 (bs, 1H). ^13^C-NMR (DMSO-*d_6_*) δ: 24.95, 37.07, 55.67, 56.43 (2C), 60.53, 89.00, 102.81 (2C), 110.62, 120.47, 127.48, 128.07, 128.46, 133.23, 135.75, 144.80, 153.59 (2C), 155.95, 157.04, 162.09, 166.69. MS (ESI): [M+1]^+^ = 450.9. Anal. calcd for C_23_H_26_N_6_O_4_. C, 61.32; H, 5.82; N, 18.66; found: C, 61.12; H, 5.58; N, 18.45.

N^2^-(3,4-Dimethoxybenzyl)-5-methyl-N^7^-(3,4,5-trimethoxyphenyl)-[1,2,4]triazolo [1,5-*a*]pyrimidine-2,7-diamine (**7v**)

Following general procedure C, using 3′,4′,5′-trimethoxyaniline as the aniline, the crude residue was purified by flash chromatography, using ethyl acetate:methanol 9.5:0.5 as eluent, to furnish **7v** as a white solid. Yield: 64%, mp 194–196 °C. ^1^H-NMR (DMSO-*d_6_*) δ: 2.29 (s, 3H), 3.66 (s, 3H), 3.69 (s, 6H), 3.71 (s, 3H), 3.76 (s, 3H), 4.14 (d, *J* = 6.4 Hz, 2H), 6.21 (s, 1H), 6.71 (s, 2H), 6.86 (d, *J* = 8.0 Hz, 1H), 6.90 (d, *J* = 8.0 Hz, 1H), 6.70–6.71 (m, 2H), 9.25 (s, 1H). ^13^C-NMR (DMSO-*d_6_*) δ: 24.96, 45.87, 55.88, 55.99, 56.44 (2C), 60.54, 88.98, 102.79 (2C), 111.95, 112.10, 119.77, 133.24, 122.46, 135.78, 144.74, 148.06, 148.99, 153.63 (2C), 155.90, 162.09, 166.49. MS (ESI): [M+1]^+^ = 481.5. Anal. calcd for C_24_H_28_N_6_O_5_. C, 59.99; H, 5.87; N, 17.49; found: C, 59.78; H, 5.69; N, 17.32.

5-Methyl-N^2^-(4-(trifluoromethyl)benzyl)-N^7^-(3,4,5-trimethoxyphenyl)-[1,2,4]triazolo [1,5-*a*]pyrimidine-2,7-diamine (**7w**)

Following general procedure C, using 3′,4′,5′-trimethoxyaniline as the aniline, the crude residue was purified by flash chromatography, using ethyl acetate/methanol 9.75:0.25 as eluent, to furnish **7w** as a white solid. Yield: 58%, mp 174–176 °C. ^1^H-NMR (DMSO-*d_6_*) δ: 2.29 (s, 3H), 3.66 (s, 3H), 3.75 (s, 6H), 4.58 (d, *J* = 6.4 Hz, 2H), 6.21 (s, 1H), 6.70 (s, 2H), 7.22 (t, *J* = 6.4 Hz, 1H), 7.57 (d, *J* = 8.0 Hz, 2H), 7.66 (d, *J* = 8.0 Hz, 2H), 9.26 (s, 1H). ^13^C-NMR (DMSO-*d_6_*) δ: 24.95, 45.64, 56.45 (2C), 60.54, 89.12, 102.88 (2C), 109.99, 125.48 (2C), 128.24 (2C), 133.17, 135.83, 138.20, 144.85, 146.17, 153.63 (2C), 155.92, 162.22, 166.36. MS (ESI): [M+1]^+^ = 489.8. Anal. calcd for C_23_H_23_F_3_N_6_O_3_. C, 56.55; H, 4.75; N, 17.21; found: C, 56.39; H, 4.58; N, 17.03.

4-(((5-Methyl-7-((3,4,5-trimethoxyphenyl)amino)-[1,2,4]triazolo [1,5-*a*]pyrimidin-2-yl)amino)methyl)benzonitrile (**7x**)

Following general procedure C, using 3′,4′,5′-trimethoxyaniline as the aniline, the crude residue was purified by flash chromatography, using ethyl acetate/methanol 9.5:0.5 as eluent, to furnish **7x** as a white solid. Yield: 78%, mp 178–180 °C. ^1^H-NMR (DMSO-*d_6_*) δ: 2.29 (s, 3H), 3.66 (s, 3H), 3.75 (s, 6H), 4.56 (d, *J* = 6.8 Hz, 2H), 6.21 (s, 1H), 6.70 (s, 2H), 7.23 (t, *J* = 6.8 Hz, 1H), 7.54 (d, *J* = 8.0 Hz, 2H), 7.63 (d, *J* = 8.0 Hz, 2H), 9.27 (s, 1H). ^13^C-NMR (DMSO-*d_6_*) δ: 24.95, 45.77, 56.45 (2C), 60.54, 89.15, 102.89 (2C), 109.76, 119.45, 128.46 (2C), 132.62 (2C), 133.15, 135.85, 144.86, 147.24, 153.63 (2C), 155.91, 162.25, 166.30. MS (ESI): [M+1]^+^ = 445.8. Anal. calcd for C_23_H_23_N_7_O_3_. C, 62.01; H, 5.20; N, 22.01; found: C, 61.88; H, 5.06; N, 21.88.

N^2^-(Benzo[d][1,3]dioxol-5-ylmethyl)-5-methyl-N^7^-(3,4,5-trimethoxyphenyl)-[1,2,4]triazolo [1,5-*a*]pyrimidine-2,7-diamine (**7y**)

Following general procedure C, using 3′,4′,5′-trimethoxyaniline as the aniline, the crude residue was purified by flash chromatography, using ethyl acetate:methanol 9.5:0.5 as eluent, to furnish **7y** as a white solid. Yield: 72%, mp 190–192 °C. ^1^H-NMR (DMSO-*d_6_*) δ: 2.29 (s, 3H), 3.66 (s, 3H), 3.76 (s, 6H), 4.38 (d, *J* = 6.4 Hz, 2H), 5.94 (s, 2H), 6.20 (s, 1H), 6.72 (s, 2H), 6.83–6.84 (m, 2H), 6.96 (s, 1H), 7.01 (t, *J* = 8.4 Hz, 1H), 9.25 (s, 1H). ^13^C-NMR (DMSO-*d_6_*) δ: 15.61, 24.95, 45.83, 56.45 (2C), 65.36, 89.00, 101.14, 102.87 (2C), 108.38, 120.84, 133.22, 135.08, 136.02, 144.80, 146.33, 147.53, 153.63 (2C), 155.90, 162.10, 166.42. MS (ESI): [M+1]^+^ = 465.4. Anal. calcd for C_23_H_24_N_6_O_5_. C, 59.48; H, 5.21; N, 18.09; found: C, 59.26; H, 5.01; N, 17.88.

5-Methyl-N^2^-phenethyl-N^7^-(3,4,5-trimethoxyphenyl)-[1,2,4]triazolo [1,5-*a*]pyrimidine-2,7-diamine (**7z**)

Following general procedure C, using 3′,4′,5′-trimethoxyaniline as the aniline, the crude residue was purified by flash chromatography, using ethyl acetate:methanol 9.75:0.25 as eluent, to furnish **7z** as a white solid. Yield: 64%, mp 201–203 °C. ^1^H-NMR (DMSO-*d_6_*) δ: 2.32 (s, 3H), 2.91 (t, *J* = 7.2 Hz, 2H), 3.49–3.51 (m, 2H), 3.68 (s, 3H), 3.78 (s, 6H), 6.22 (s, 1H), 6.63 (t, *J* = 6.4 Hz, 1H), 6.74 (s, 2H), 7.18–7.20 (m, 1H), 7.29–7.31 (m, 4H), 9.26 (bs, 1H).^13^C-NMR (DMSO-*d_6_*) δ: 24.40, 35.27, 43.92, 55.89 (2C), 59.98, 88.35, 102.31 (2C), 125.86, 128.15 (2C), 128.64 (2C), 132.68, 135.23, 139.74, 144.22, 153.07 (2C), 155.32, 161.49, 165.78. MS (ESI): [M+1]^+^ = 434.8. Anal. calcd for C_23_H_26_N_6_O_3_. C, 63.58; H, 6.03; N, 19.34; found: C, 63.41; H, 5.90; N, 19.22.

N^2^-(4-Fluorophenethyl)-5-methyl-N^7^-(3,4,5-trimethoxyphenyl)-[1,2,4]triazolo [1,5-*a*]pyrimidine-2,7-diamine (**7aa**)

Following general procedure C, using 3′,4′,5′-trimethoxyaniline as the aniline, the crude residue was purified by flash chromatography, using ethyl acetate:methanol 9.75:0.25 as eluent, to furnish **7aa** as a white solid. Yield: 64%, mp 150–152 °C. ^1^H-NMR (DMSO-*d_6_*) δ: 2.30 (s, 3H), 2.88 (t, *J* = 7.2 Hz, 2H), 3.44–3.49 (m, 2H), 3.66 (s, 3H), 3.76 (s, 6H), 6.20 (s, 1H), 6.62 (t, *J* = 6.4 Hz, 1H), 6.72 (s, 2H), 7.10 (t, *J* = 8.4 Hz, 2H), 7.26–7.29 (m, 2H), 9.23 (bs, 1H). ^13^C-NMR (DMSO-*d_6_*) δ: 24.96, 34.87, 44.46, 56.46 (2C), 60.55, 88.92, 102.89 (2C), 115.24 and 115.46 (*J*_2CF_ = 21.4 Hz, 2C), 130.94 and 131.00 (*J*_3CF_ = 6.0 Hz, 2C), 133.24, 135.81, 136.47, 144.80, 153.63 (2C), 155.87, 162.07, 162.43, 166.32. MS (ESI): [M+1]^+^ = 453.5. Anal. calcd for C_23_H_25_FN_6_O_3_. C, 61.05; H, 5.57; N, 18.57; found: C, 60.89; H, 5.42; N, 18.39.

N^2^-(4-Chlorophenethyl)-5-methyl-N^7^-(3,4,5-trimethoxyphenyl)-[1,2,4]triazolo [1,5-*a*]pyrimidine-2,7-diamine (**7ab**)

Following general procedure C, using 3′,4′,5′-trimethoxyaniline as the aniline, the crude residue was purified by flash chromatography, using ethyl acetate:methanol 9.75:0.25 as eluent, to furnish **7ab** as a white solid. Yield: 63%, mp 196–198 °C. ^1^H-NMR (DMSO-*d_6_*) δ: 2.32 (s, 3H), 2.89 (t, *J* = 7.2 Hz, 2H), 3.48–3.50 (m, 2H), 3.68 (s, 3H), 3.78 (s, 6H), 6.22 (s, 1H), 6.65 (t, *J* = 6.4 Hz, 1H), 6.74 (s, 2H), 7.29 (d, *J* = 8.4 Hz, 2H), 7.34 (d, *J* = 8.4 Hz, 2H), 9.25 (bs, 1H). ^13^C-NMR (DMSO-*d_6_*) δ: 24.38, 34.41, 43.63, 55.88 (2C), 59.98, 88.35, 102.32 (2C), 128.02 (2C), 130.48 (2C), 130.55, 132.66, 135.24, 138.82, 144.22, 153.06 (2C), 155.29, 161.50, 165.72. MS (ESI): [M+1]^+^ = 469.5. Anal. calcd for C_23_H_25_ClN_6_O_3_. C, 58.91; H, 5.37; N, 17.92; found: C, 58.69; H, 5.22; N, 17.71.

N^2^-(4-Methoxyphenethyl)-5-methyl-N^7^-(3,4,5-trimethoxyphenyl)-[1,2,4]triazolo [1,5-*a*]pyrimidine-2,7-diamine (**7ac**)

Following general procedure C, using 3′,4′,5′-trimethoxyaniline as the aniline, the crude residue was purified by flash chromatography, using ethyl acetate:methanol 9.75:0.25 as eluent, to furnish **7ac** as a white solid. Yield: 63%, mp 214–216 °C. ^1^H-NMR (DMSO-*d_6_*) δ: 2.30 (s, 3H), 2.84 (t, *J* = 7.2 Hz, 2H), 3.40–3.45 (m, 2H), 3.66 (s, 3H), 3.70 (s, 3H), 3.76 (s, 6H), 6.20 (s, 1H), 6.72 (t, *J* = 6.4 Hz, 1H), 6.74 (s, 2H), 6.83 (d, *J* = 8.8 Hz, 2H), 7.16 (d, *J* = 8.8 Hz, 2H), 9.22 (bs, 1H). ^13^C-NMR (DMSO-*d_6_*) δ: 24.95, 34.95, 44.72, 55.42, 56.44 (2C), 60.54, 88.90, 102.83 (2C), 114.14 (2C), 130.14 (2C), 132.16, 133.25, 135.75, 144.77, 153.62 (2C), 155.87, 158.05, 162.03, 166.35. MS (ESI): [M+1]^+^ = 464.8. Anal. calcd for C_24_H_28_N_6_O_4_. C, 62.06; H, 6.08; N, 18.09; found: C, 61.88; H, 5.89; N, 17.91.

N^2^-(3-Phenylpropyl)-N^7^-(3,4,5-trimethoxyphenyl)-[1,2,4]triazolo [1,5-*a*]pyrimidine-2,7-diamine (**7ad**)

Following general procedure C, using 3′,4′,5′-trimethoxyaniline as the aniline, the crude residue was purified by flash chromatography, using ethyl acetate:methanol 9.75:0.25 as eluent, to furnish **7ad** as a white solid. Yield: 64%, mp 201–203 °C. ^1^H-NMR (DMSO-*d_6_*) δ: 1.82–1.88 (m, 2H), 2.24 (s, 3H), 2.65 (t, *J* = 7.2 Hz, 2H), 3.22–3.24 (m, 2H), 3.66 (s, 3H), 3.76 (s, 6H), 6.20 (s, 1H), 6.62 (t, *J* = 6.4 Hz, 1H), 6.72 (s, 2H), 7.20 (t, *J* = 6.4 Hz, 1H), 7.22–7.26 (m, 4H), 9.19 (bs, 1H). ^13^C-NMR (DMSO-*d_6_*) δ: 24.95, 31.60, 33.10, 42.34, 56.45 (2C), 60.54, 88.89, 102.77 (2C), 126.13 (2C), 128.73 (2C), 133.28, 135.75, 142.42, 144.74, 153.62 (2C), 154.32, 155.88, 162.00, 166.56. MS (ESI): [M+1]^+^ = 449.4. Anal. calcd for C_24_H_28_N_6_O_3_. C, 62.06; H, 6.08; N, 18.09; found: C, 61.88; H, 5.96; N, 17.89.

N^7^-(3-Chloro-4-fluorophenyl)-N^2^-(4-chlorobenzyl)-5-methyl-[1,2,4]triazolo [1,5-*a*]pyrimidine-2,7-diamine (**8a**)

Following general procedure C, the crude residue obtained by condensation of **11n** and 3′-chloro,4′-fluoroaniline was purified by flash chromatography, using ethyl acetate:methanol 9.5:0.5 as eluent, to furnish **8a** as a white solid. Yield: 68%, mp 186–188 °C. ^1^H-NMR (DMSO-*d_6_*) δ: 2.48 (s, 3H), 4.57 (bs, 2H), 5.95 (s, 2H), 6.18 (s, 1H), 6.40 (bs, 1H), 7.23–7.26 (m, 2H), 7.28 (d, *J* = 8.8 Hz, 2H), 7.30 (d, *J* = 8.8 Hz, 1H), 7.84 (bs, 1H). ^13^C-NMR (DMSO-*d_6_*) δ: 24.99, 45.93, 90.50, 117.79, 118.01, 124.71, 124.78, 127.20, 128.71 (2C), 131.75, 133.21, 136.73, 144.45, 151.38, 155.33, 156.28, 164.73, 165.63. MS (ESI): [M]^+^ = 417.3, [M + 2] ^+^ = 419.3. Anal. calcd for C_19_H_15_Cl_2_FN_6_. C, 54.69; H, 3.62; N, 20.14; found: C, 54.56; H, 3.48; N, 20.01.

N^7^-(3-Chlorophenyl)-N^2^-(4-chlorobenzyl)-5-methyl-[1,2,4]triazolo [1,5-*a*]pyrimidine-2,7-diamine (**8b**)

Following general procedure C, the crude residue obtained by condensation of **11n** and 3′-chloroaniline was purified by flash chromatography, using ethyl acetate as eluent, to furnish **8b** as a white solid. Yield: 74%, mp 158–160 °C. ^1^H-NMR (DMSO-*d_6_*) δ: 2.30 (s, 3H), 4.47 (d, *J* = 6.4 Hz, 2H), 6.24 (s, 1H), 7.17 (t, *J* = 6.4 Hz, 1H), 7.32 (dd, *J* = 8.0 and 2.4 Hz, 1H), 7.35–7.40 (m, 4H), 7.42 (d, *J* = 8.8 Hz, 2H), 7.45 (s, 1H), 9.54 (s, 1H). ^13^C-NMR (DMSO-*d_6_*) δ: 24.96, 45.36, 89.32, 122.83, 124.17, 125.71, 128.54 (2C), 129.50 (2C), 131.38, 131.55, 133.99, 139.39, 140.18, 144.14, 155.94, 162.32, 166.46. MS (ESI): [M]^+^ = 399.0. Anal. calcd for C_19_H_16_Cl_2_N_6_. C, 57.15; H, 4.04; N, 21.05; found: C, 56.94; H, 3.89; N, 20.91.

N^7^-(3-Ethynylphenyl)-N^2^-(4-chlorobenzyl)-5-methyl-[1,2,4]triazolo [1,5-*a*]pyrimidine-2,7-diamine (**8c**)

Following general procedure C, the crude residue obtained by condensation of **11n** and 3′-ethynylaniline was purified by flash chromatography, using ethyl acetate as eluent, to furnish **8c** as a white solid. Yield: 82%, mp 190–192 °C. ^1^H-NMR (DMSO-*d_6_*) δ: 2.29 (s, 3H), 4.25 (s, 1H), 4.47 (d, *J* = 6.8 Hz, 2H), 6.17 (s, 1H), 7.16 (t, *J* = 6.8 Hz, 1H), 7.34–7.38 (m, 5H), 7.40–7.46 (m, 3H), 9.51 (s, 1H). ^13^C-NMR (DMSO-*d_6_*) δ: 24.96, 45.36, 81.94, 83.23, 89.07, 106.08, 123.24, 125.24, 127.53, 128.54, 129.27, 129.52, 130.29, 131.54, 138.06, 140.20, 144.37, 155.95, 156.31, 162.25, 166.44. MS (ESI): [M]^+^ = 389.3. Anal. calcd for C_21_H_17_ClN_6_. C, 64.86; H, 4.41; N, 21.61; found: C, 64.68; H, 4.25; N, 21.44.

N^7^-(3-Chloro-4-fluorophenyl)-N^2^-(4-methylbenzyl)-5-methyl-[1,2,4]triazolo [1,5-*a*]pyrimidine-2,7-diamine (**8d**)

Following general procedure C, the crude residue obtained by condensation of **11p** and 3′-chloro,4′-fluoroaniline was purified by flash chromatography, using ethyl acetate:methanol 9.5:0.5 as eluent, to furnish **8d** as a white solid. Yield: 70%, mp 178–180 °C. ^1^H-NMR (DMSO-*d_6_*) δ: 2.26 (s, 3H), 2.35 (s, 3H), 4.46 (d, *J* = 6.8 Hz, 2H), 6.15 (s, 1H), 7.07 (t, *J* = 6.8 Hz, 1H), 7.10 (d, *J* = 8.0 Hz, 2H), 7.25 (d, *J* = 8.0 Hz, 2H), 7.38–7.42 (m, 1H), 7.47 (t, *J* = 8.8 Hz, 1H), 7.62 (dd, *J* = 8.8 and 2.4 Hz, 1H), 9.52 (bs, 1H). ^13^C-NMR (DMSO-*d_6_*) δ: 24.34, 28.88, 45.18, 88.41, 117.20 and 117.42 (*J*_1CF_ = 22 Hz, 1C), 119.86, 125.09, 126.52, 127.00 (2C), 128.56, 134.44, 135.40, 137.45, 143.98, 152.32, 153.62 and 156.06 (*J*_1CF_ = 243 Hz, 1C), 155.33, 161.61, 166.04. MS (ESI): [M]^+^ =417.3, [M + 2] ^+^ = 397.2. Anal. calcd for C_20_H_18_ClFN_6_. C, 60.53; H, 4.57; N, 21.18; found: C, 60.38; H, 4.44; N, 21.02.

N^7^-(3-Chlorophenyl)-N^2^-(4-methylbenzyl)-5-methyl-[1,2,4]triazolo [1,5-*a*]pyrimidine-2,7-diamine (**8e**)

Following general procedure C, the crude residue obtained by condensation of **11p** and 3′-chloroaniline was purified by flash chromatography, using ethyl acetate as eluent, to furnish **8e** as a white solid. Yield: 78%, mp 166–168 °C. ^1^H-NMR (DMSO-*d_6_*) δ: 2.26 (s, 3H), 2.32 (s, 3H), 4.47 (d, *J* = 6.4 Hz, 2H), 6.25 (s, 1H), 7.11 (t, *J* = 6.4 Hz, 1H), 7.12 (d, *J* = 8.0 Hz, 1H), 7.26 (d, *J* = 8.0 Hz, 2H), 7.29 (d, *J* = 8.0 Hz, 1H), 7.36–7.40 (m, 2H), 7.44 (t, *J* = 8.0 Hz, 2H), 9.56 (s, 1H). ^13^C-NMR (DMSO-*d_6_*) δ: 20.55, 24.39, 28.89, 45.19, 88.67, 122.21, 123.55, 125.08, 127.03 (2C), 128.56 (2C), 130.80, 133.42, 135.40, 137.45, 138.86, 143.52, 161.64, 166.04. MS (ESI): [M]^+^ = 379.1. Anal. calcd for C_20_H_19_ClN_6_. C, 63.40; H, 5.05; N, 22.18; found: C, 63.23; H, 4.91; N, 22.02.

N^2^-(Benzo[d][1,3]dioxol-5-ylmethyl)-N^7^-(3-chloro-4-fluorophenyl)-5-methyl-[1,2,4]triazolo [1,5-*a*]pyrimidine-2,7-diamine (**8f**)

Following general procedure C, the crude residue was purified by flash chromatography, using ethyl acetate:methanol 9.5:0.5 as eluent, to furnish **8f** as a white solid. Yield: 68%, mp 186–188 °C. ^1^H-NMR (DMSO-*d_6_*) δ: 2.31 (s, 3H), 4.40 (d, *J* = 6.4 Hz, 2H), 5.95 (s, 2H), 6.15 (s, 1H), 6.84–6.85 (m, 2H), 6.96 (s, 1H), 7.07 (t, *J* = 6.4 Hz, 1H), 7.44–7.50 (m, 2H), 7.64 (dd, *J* = 6.8 and 2.4 Hz, 1H), 9.54 (s, 1H). ^1^^3^C-NMR (DMSO-*d_6_*) δ: 24.36, 45.26, 88.44, 100.58, 107.81, 117.23 and 117.45 (*J*_2CF_ = 22 Hz, 1C), 119.69, 119.87, 120.23, 125.15, 125.22, 126.58, 134.46, 144.00, 145.76, 146.98, 153.66 and 155.33 (*J*_1CF_ = 233 Hz, 1C), 156.10, 161.67, 165.93. MS (ESI): [M+1]^+^ = 427.3. Anal. calcd for C_20_H_16_ClFN_6_O_2_. C, 56.28; H, 3.78; N, 19.69; found: C, 56.03; H, 3.65; N, 19.58.

N^2^-(Benzo[d][1,3]dioxol-5-ylmethyl)-N^7^-(4-fluorophenyl)-5-methyl-[1,2,4]triazolo [1,5-*a*]pyrimidine-2,7-diamine (**8g**)

Following general procedure C, the crude residue obtained by condensation of **11y** and 4′-fluoroaniline was purified by flash chromatography, using ethyl acetate as eluent, to furnish **8g** as a white solid. Yield: 75%, mp 168–170 °C. ^1^H-NMR (DMSO-*d_6_*) δ: 2.28 (s, 3H), 4.40 (d, *J* = 6.4 Hz, 2H), 5.96 (s, 2H), 6.04 (s, 1H), 6.84–6.86 (m, 2H), 6.97 (s, 1H), 7.04 (t, *J* = 6.4 Hz, 1H), 7.27 (t, *J* = 9.2 Hz, 2H), 7.42 (dd, *J* = 9.2 and 4.8 Hz, 2H), 9.45 (s, 1H). ^13^C-NMR (DMSO-*d_6_*) δ: 24.35, 45.25, 87.96, 100.57, 107.80 and 107.86 (*J*_3CF_ = 6 Hz, 2C), 115.91 and 116.14 (*J*_2CF_ = 23 Hz, 2C), 120.26, 126.78, 126.86, 133.26, 134.48, 144.40 and 146.95 (*J*_1CF_ = 255 Hz, 1C), 145.75, 155.35, 160.97, 161.48, 162.86, 165.89. MS (ESI): [M+1]^+^ = 393.1. Anal. calcd for C_20_H_17_FN_6_O_2_. C, 61.22; H, 4.37; N, 21.42; found: C, 61.03; H, 4.23; N, 21.19.

N^2^-(Benzo[d][1,3]dioxol-5-ylmethyl)-N^7^-(3-fluorophenyl)-5-methyl-[1,2,4]triazolo [1,5-*a*]pyrimidine-2,7-diamine (**8h**)

Following general procedure C, the crude residue obtained by condensation of **11y** and 3′-fluoroaniline was purified by flash chromatography, using ethyl acetate as eluent, to furnish **8h** as a white solid. Yield: 63%, mp 174–176 °C. ^1^H-NMR (CDCl_3_) δ: 2.50 (s, 3H), 4.50 (d, J = 4.4 Hz, 2H), 5.92 (s, 2H), 6.34 (s, 1H), 6.73–6.75 (m, 2H), 6.84 (d, *J* = 9.2 Hz, 1H), 6.86 (s, 1H), 7.23 (d, *J* = 8.0 Hz, 1H), 7.38 (t, *J* = 7.6 Hz, 1H), 7.46–7.53 (m, 2H), 7.90 (s, 1H). ^13^C-NMR (CDCl_3_) δ: 25.16, 46.63, 90.56, 101.15, 108.18, 108.37, 111.15 and 111.39 (*J*_2CF_ = 24 Hz, 1C), 114.02 and 114.23 (*J*_2CF_ = 21 Hz, 1C), 119.47, 120.80, 131.41 and 131.50 (*J*_3CF_ = 9 Hz, 1C), 132.22, 137.07 and 137.17 (*J*_3CF_ = 9 Hz, 1C), 143.89, 147.05, 147.96, 152.03, 158.08, 162.63, 162.16 and 164.32 (*J*_1CF_ = 216 Hz, 1C). MS (ESI): [M+1]^+^ = 393.2. Anal. calcd for C_20_H_17_FN_6_O_2_. C, 61.22; H, 4.37; N, 21.42; found: C, 61.10; H, 4.25; N, 21.22.

N^2^-(Benzo[d][1,3]dioxol-5-ylmethyl)-N^7^-(4-chlorophenyl)-5-methyl-[1,2,4]triazolo [1,5-*a*]pyrimidine-2,7-diamine (**8i**)

Following general procedure C, the crude residue obtained by condensation of **11y** and 4′-chloroaniline was purified by flash chromatography, using ethyl acetate:methanol 9.5:0.5 as eluent, to furnish **8i** as a gray solid. Yield: 71%, mp 156–158 °C. ^1^H-NMR (DMSO-*d_6_*) δ: 2.28 (s, 3H), 4.39 (d, *J* = 6.4 Hz, 2H), 5.94 (s, 2H), 6.17 (s, 1H), 6.82–6.83 (m, 2H), 6.95 (s, 1H), 7.03 (t, *J* = 6.4 Hz, 1H), 7.40 (d, *J* = 9.2 Hz, 2H), 7.47 (d, *J* = 9.2 Hz, 2H), 9.52 (s, 1H). ^13^C-NMR (DMSO-*d_6_*) δ: 24.35, 45.26, 88.40, 100.57, 107.80, 107.83, 120.24, 125.79 (2C), 129.18 (2C), 129.39, 134.45, 136.15, 143.78, 145.75, 146.97, 153.36, 161.59, 165.91. MS (ESI): [M+1]^+^ = 409.2. Anal. calcd for C_20_H_17_ClN_6_O_2_. C, 58.75; H, 4.19; N, 20.56; found: C, 58.61; H, 4.04; N, 20.39.

N^2^-(Benzo[d][1,3]dioxol-5-ylmethyl)-N^7^-(3-chlorophenyl)-5-methyl-[1,2,4]triazolo [1,5-*a*]pyrimidine-2,7-diamine (**8j**)

Following general procedure C, the crude residue obtained by condensation of **11y** and 3′-chloroaniline was purified by flash chromatography, using ethyl acetate as eluent, to furnish **8j** as a white solid. Yield: 71%, mp 164–166 °C. ^1^H-NMR (CDCl_3_) δ: 2.49 (s, 3H), 4.50 (d, *J* = 3.2 Hz, 2H), 5.93 (s, 2H), 6.34 (s, 1H), 6.46 (bs, 1H), 6.73 (d, *J* = 8.0 Hz, 1H), 6.82 (d, *J* = 9.2 Hz, 1H), 6.90 (s, 1H), 7.28 (d, *J* = 8.0 Hz, 1H), 7.34–7.37 (m, 2H), 7.41 (t, *J* = 7.6 Hz, 1H), 7.86 (s, 1H). ^13^C-NMR (CDCl_3_) δ: 25.01, 46.43, 91.07, 101.06, 108.06, 108.28, 120.72, 122.21, 124.29, 127.56, 131.10, 131.77, 135.74, 136.38, 139.61, 144.06, 147.01, 147.87, 161.24, 164.97. MS (ESI): [M+1]^+^ = 409.2. Anal. calcd for C_20_H_17_ClN_6_O_2_. C, 58.75; H, 4.19; N, 20.56; found: C, 58.58; H, 4.01; N, 20.33.

N^2^-(Benzo[d][1,3]dioxol-5-ylmethyl)-N^7^-(3,4-dichlorophenyl)-5-methyl-[1,2,4]triazolo [1,5-*a*]pyrimidine-2,7-diamine (**8k**)

Following general procedure C, the crude residue obtained by condensation of **11y** and 3′,4′-dichlorolaniline was purified by flash chromatography, using ethyl acetate:petroleum ether 9:1 as eluent, to furnish **8k** as a white solid. Yield: 72%, mp 166–168 °C. ^1^H-NMR (DMSO-*d_6_*) δ: 2.50 (s, 3H), 4.48 (bs, 2H), 5.93 (s, 2H), 6.34 (s, 1H), 6.71 (d, *J* = 7.6 Hz, 1H), 6.81 (d, *J* = 8.4 Hz, 1H), 6.84 (s, 1H), 7.12 (bs, 1H), 7.26 (dd, *J* = 8.4 and 2.4 Hz, 1H), 7.50 (d, *J* = 2.4 Hz, 1H), 7.56 (d, *J* = 8.4 Hz, 1H), 8.14 (s, 1H). ^13^C-NMR (DMSO-*d_6_*) δ: 24.91, 65.84, 92.08, 101.10, 108.00, 108.28, 120.73, 123.87, 126.42, 130.25, 131.23, 131.79. 134.15, 140.51, 144.31, 147.07, 147.88, 148.57, 159.24, 165.66. MS (ESI): [M+1]^+^ = 443.3. Anal. calcd for C_20_H_16_Cl_2_N_6_O_2_. C, 54.19; H, 3.64; N, 18.96; found: C, 54.01; H, 3.49; N, 18.77.

N^2^-(Benzo[d][1,3]dioxol-5-ylmethyl)-N^7^-(4-bromophenyl)-5-methyl-[1,2,4]triazolo [1,5-*a*]pyrimidine-2,7-diamine (**8l**)

Following general procedure C, the crude residue obtained by condensation of **11y** and 3′-bromoaniline was purified by flash chromatography, using ethyl acetate as eluent, to furnish **8l** as a white solid. Yield: 80%, mp 182–184 °C. ^1^H-NMR (DMSO-*d_6_*) δ: 2.47 (s, 3H), 4.50 (bs, 2H), 5.93 (s, 2H), 6.26 (s, 1H), 6.38 (bs, 1H), 6.72 (d, *J* = 8.0 Hz, 1H), 6.84 (d, *J* = 8.0 Hz, 1H), 6.87 (s, 1H), 7.23 (d. *J* = 8.4 Hz, 2H), 7.60 (d, *J* = 8.4 Hz, 2H), 7.83 (bs, 1H). ^13^C-NMR (DMSO-*d_6_*) δ: 24.96, 49.46, 90.67, 91.41, 101.05, 108.06, 108.27, 120.70, 125.87 (2C), 131.94, 133.22 (2C), 134.32, 144.19, 146.98, 147.86, 159.36, 161.77, 164.63. MS (ESI): [M+1]^+^ = 453.0, 455.2. Anal. calcd for C_20_H_17_BrN_6_O_2_. C, 52.99; H, 3.78; N, 18.54; found: C, 52.77; H, 3.63; N, 18.36.

N^2^-(Benzo[d][1,3]dioxol-5-ylmethyl)-N^7^-(3-bromophenyl)-5-methyl-[1,2,4]triazolo [1,5-*a*]pyrimidine-2,7-diamine (**8m**)

Following general procedure C, the crude residue obtained by condensation of **11y** and 3′-bromoaniline was purified by flash chromatography, using ethyl acetate as eluent, to furnish **8m** as a gray solid. Yield: 58%, mp 150–152 °C. ^1^H-NMR (CDCl_3_) δ: 2.50 (s, 3H), 4.51 (bs, 2H), 5.93 (s, 2H), 6.27 (s, 1H), 6.74–6.76 (m, 1H), 6.83–6.88 (m, 2H), 7.32–7.35 (m, 3H), 7.46 (t, *J* = 7.6 Hz, 1H), 7.53 (s, 1H), 7.98 (bs, 1H). ^13^C-NMR (CDCl_3_) δ: 25.15, 46.66, 90.11, 101.12, 108.19, 108.32, 120.79, 122.56, 123.46, 127.00, 130.07, 131.28, 132.44, 137.14, 143.90, 144.12, 146.99, 147.64, 161.32, 164.25. MS (ESI): [M]^+^ = 453.2, 455.2. Anal. calcd for C_20_H_17_BrN_6_O_2_. C, 52.99; H, 3.78; N, 18.54; found: C, 52.80; H, 3.66; N, 18.40.

N^2^-(Benzo[d][1,3]dioxol-5-ylmethyl)-N^7^-(4-trifluoromethylphenyl)-5-methyl-[1,2,4]triazolo [1,5-*a*]pyrimidine-2,7-diamine (**8n**)

Following general procedure C, the crude residue obtained by condensation of **11y** and 4′-trifluoromethylaniline was purified by flash chromatography, using ethyl acetate as eluent, to furnish **8n** as a white solid. Yield: 58%, mp 158–160 °C. ^1^H-NMR (DMSO-*d_6_*) δ: 2.51 (s, 3H), 4.53 (d, *J* = 6.2 Hz, 2H), 5.94 (s, 2H), 6.39 (s, 1H), 6.75 (d, *J* = 8.0 Hz, 1H), 6.84 (dd, *J* = 8.0 and 1.0 Hz, 1H), 6.89 (d, *J* = 1.0 Hz, 1H), 7.05 (t, *J* = 6.2 Hz, 1H), 7.47 (d, *J* = 8.4 Hz, 2H), 7.74 (d, *J* = 8.4 Hz, 2H), 9.50 (s, 1H). ^13^C-NMR (DMSO-*d_6_*) δ: 24.89, 45.78, 88.55, 100.88, 107.80, 120.22, 122.51 and 124.87 (J_1CF_ = 236 Hz, 1C), 125.86 (2C), 129.82 (2C), 129.55, 134.55, 136.73, 144.12, 146.12, 146.97, 153.50, 161.59, 166.15. MS (ESI): [M+1]^+^ = 443.4. Anal. calcd for C_21_H_17_F_3_N_6_O_2_. C, 57.01; H, 3.87; N, 19.00; found: C, 56.88; H, 3.73; N, 18.88.

N^2^-(Benzo[d][1,3]dioxol-5-ylmethyl)-N^7^-(3-trifluoromethylphenyl)-5-methyl-[1,2,4]triazolo [1,5-*a*]pyrimidine-2,7-diamine (**8o**)

Following general procedure C, the crude residue obtained by condensation of **11y** and 3′-trifluoromethylaniline was purified by flash chromatography, using ethyl acetate as eluent, to furnish **8o** as a white solid. Yield: 64%, mp 180–182 °C. ^1^H-NMR (DMSO-*d_6_*) δ: 2.32 (s, 3H), 4.41 (d, *J* = 6.4 Hz, 2H), 5.96 (s, 2H), 6.27 (s, 1H), 6.85–6.86 (m, 2H), 6.97 (s, 1H), 7.09 (t, *J* = 6.4 Hz, 1H), 7.60 (d, *J* = 8.8 Hz, 1H), 7.68 (t, *J* = 8.8 Hz, 1H), 7.74 (d, *J* = 8.8 Hz, 1H), 7.76 (s, 1H), 9.70 (s, 1H). ^13^C-NMR (DMSO-*d_6_*) δ: 25.06, 45.92, 89.31, 101.24, 108.48 (2C), 120.92, 121.10, 122.28, 123.09 and 125.79 (*J*_1CF_ = 271 Hz, 1C), 127.99, 130.51 and 130.82 (*J*_2CF_ = 31 Hz, 1C), 131.11, 135.10, 138.92, 144.18, 146.43, 147.64, 156.06, 162.39. MS (ESI): [M+1]^+^ = 443.3. Anal. calcd for C_21_H_17_F_3_N_6_O_2_. C, 57.01; H, 3.87; N, 19.00; found: C, 56.78; H, 3.71; N, 18.80.

N^2^-(Benzo[d][1,3]dioxol-5-ylmethyl)-N^7^-(3-ethynylphenyl)-5-methyl-[1,2,4]triazolo [1,5-*a*]pyrimidine-2,7-diamine (**8p**)

Following general procedure C, the crude residue obtained by condensation of **11y** and 3′-ethynylaniline was purified by flash chromatography, using ethyl acetate as eluent, to furnish **8p** as a white solid. Yield: 78%, mp 165–168 °C. ^1^H-NMR (DMSO-*d_6_*) δ: 2.29 (s, 3H), 4.25 (s, 1H), 4.39 (d, *J* = 6.0 Hz, 2H), 5.94 (s, 2H), 6.17 (s, 1H), 6.82 (d, *J* = 7.6 Hz, 1H), 6.84 (d, *J* = 7.6 Hz, 1H), 6.95 (s, 1H), 7.04 (t, *J* = 6.4 Hz, 1H), 7.33–7.35 (m, 1H), 7.44–7.47 (m, 3H), 9.50 (s, 1H). ^13^C-NMR (CDCl_3_) δ: 24.95, 45.83, 81.94, 83.33, 89.01, 101.14, 104.38, 104.41, 120.83, 123.24, 125.24, 127.54, 129.25, 130.29, 135.04, 138.09, 144.34, 146.34, 147.54, 155.93, 162.17, 166.46. MS (ESI): [M+1]^+^ = 399.3. Anal. calcd for C_22_H_18_N_6_O_2_. C, 66.32; H, 4.55; N, 21.09; found: C, 66.10; H, 4.39; N, 20.98.

N^2^-(3-Phenylpropyl)-N^7^-(4-fluorophenyl)-[1,2,4]triazolo [1,5-*a*]pyrimidine-2,7-diamine (**8q**)

Following general procedure C, the crude residue obtained by condensation of **11ad** and 4′-fluoroaniline was purified by flash chromatography, using ethyl acetate as eluent, to furnish **8q** as a white solid. Yield: 74%, mp 182–184 °C. ^1^H-NMR (DMSO-*d_6_*) δ: 1.83–1.88 (m, 2H), 2.26 (s, 3H), 2.65 (t, *J* = 8.0 Hz, 2H), 3.26–3.30 (m, 2H), 6.01 (s, 1H), 6.63 (t, *J* = 8.0 Hz, 1H), 7.15 (t, *J* = 6.8 Hz, 1H), 7.20 (d, *J* = 6.8 Hz, 1H), 7.22–7.28 (m, 5H), 7.40–7.44 (m, 2H), 9.41 (s, 1H). ^13^C-NMR (DMSO-*d_6_*) δ: 24.91, 31.61, 33.10, 42.34, 88.42, 116.47 and 116.70 (J_2CF_ = 23 Hz, 2C), 126.13 (2C), 127.28 and 127.36 (*J*_3CF_ = 8.4 Hz, 2C), 128.73, 128.77, 133.88, 142.41, 144.93, 155.90, 159.18 and 161.48 (*J*_1CF_ = 230 Hz, 1C), 159.26, 161.95, 166.59. MS (ESI): [M+1]^+^ = 377.2. Anal. calcd for C_21_H_21_FN_6_. C, 67.00; H, 5.62; N, 22.33; found: C, 66.89; H, 5.46; N, 22.18.

N^7^-(3-Chloro-4-fluorophenyl)-5-methyl-N^2^-(3-phenylpropyl)-[1,2,4]triazolo [1,5-*a*]pyrimidine-2,7-diamine (**8r**)

Following general procedure C, the crude residue obtained by condensation of **11ad** and 3′-chloro,4′-fluoroaniline was purified by flash chromatography, using ethyl acetate as eluent, to furnish **8r** as a white solid. Yield: 74%, mp 168–170 °C. ^1^H-NMR (DMSO-*d_6_*) δ: 1.87–1.91 (m, 2H), 2.31 (s, 3H), 2.65 (t, *J* = 8,0 Hz, 2H), 3.27–3.29 (m, 2H), 6.14 (s, 1H), 6.65 (t, *J* = 6.4 Hz, 1H), 7.17 (t, *J* = 7.2 Hz, 1H), 7.21 (d, *J* = 6.4 Hz, 1H), 7.26–7.30 (m, 2H), 7.42–7.44 (m, 2H), 7.44–7.49 (m, 1H), 7.63 (dd, *J* = 6.4 and 2.4 Hz, 1H), 9.49 (bs, 1H). ^13^C-NMR (DMSO-*d_6_*) δ: 24.36, 28.87, 31.02, 32.52, 41.78, 88.32, 117.21 and 117.42 (*J*_2CF_ = 21 Hz, 1C), 119.67 and 119.86 (*J*_2CF_ = 18 Hz, 1C), 125.07, 125.15, 125.57, 126.50, 128.16 (2C), 134.45, 141.82, 143.96, 153.62 and 156.06 (*J*_1CF_ = 244 Hz, 1C), 155.32, 161.56, 166.06. MS (ESI): [M+1]^+^ = 411.2. Anal. calcd for C_21_H_20_ClFN_6_. C, 61.39; H, 4.91; N, 20.45; found: C, 61.28; H, 4.77; N, 20.34.

N^2^-(3-Phenylpropyl)-N^7^-(4-chlorophenyl)-[1,2,4]triazolo [1,5-*a*]pyrimidine-2,7-diamine (**8s**)

Following general procedure C, the crude residue obtained by condensation of **11ad** and 4′-chloroaniline was purified by flash chromatography, using ethyl acetate as eluent, to furnish **8s** as a white solid. Yield: 70%, mp 164–166 °C. ^1^H-NMR (DMSO-*d_6_*) δ: 1.82–1.86 (m, 2H), 2.29 (s, 3H), 2.65 (t, *J* = 6.4 Hz, 2H), 3.27–3.30 (m, 2H), 6.16 (s, 1H), 6.62 (t, *J* = 6.4 Hz, 1H), 7.14 (t, *J* = 7.2 Hz, 1H), 7.22–7.26 (m, 4H), 7.41 (d, *J* = 8.8 Hz, 2H), 7.46 (d, *J* = 8.8 Hz, 2H), 9.47 (bs, 1H). ^13^C-NMR (DMSO-*d_6_*) δ: 24.92, 31.61, 33.09, 42.34, 88.87, 126.13 (2C), 126.28 (2C), 128.72 (2C), 128.76, 129.74 (2C), 129.88, 136.79, 142.40, 144.33, 155.93, 162.06, 166.62. MS (ESI): [M+1]^+^ = 393.3. Anal. calcd for C_21_H_21_ClN_6_. C, 64.20; H, 5.39; N, 21.39; found: C, 64.09; H, 5.26; N, 21.25.

N^2^-(3-Phenylpropyl)-N^7^-(3-chlorophenyl)-[1,2,4]triazolo [1,5-*a*]pyrimidine-2,7-diamine (**8t**)

Following general procedure C, the crude residue obtained by condensation of **11ad** and 3′-chloroaniline was purified by flash chromatography, using ethyl acetate as eluent, to furnish **8t** as a white solid. Yield: 72%, mp 212–214 °C. ^1^H-NMR (DMSO-*d_6_*) δ: 1.83–1.91 (m, 2H), 2.32 (s, 3H), 2.67 (t, *J* = 7.6 Hz, 2H), 3.26–3.30 (m, 2H), 6.24 (s, 1H), 6.65 (t, *J* = 6.4 Hz, 1H), 7.17 (t, *J* = 8.4 Hz, 1H), 7.22 (d, *J* = 8.0 Hz, 2H), 7.26–7.30 (m, 3H), 7.36 (d, *J* = 8.0 Hz, 1H), 7.42–7.49 (m, 1H), 7.50 (s, 1H), 9.54 (bs, 1H). ^13^C-NMR (DMSO-*d_6_*) δ: 24.38, 31.02, 32.51, 41.76, 88.59, 91.48, 122.17, 123.51 (2C), 125.04, 125.55, 128.14 (2C), 130.80, 133.41, 138.89, 141.82, 143.49, 155.35, 161.57, 166.04. MS (ESI): [M+1]^+^ = 393.2. Anal. calcd for C_21_H_21_ClN_6_. C, 64.20; H, 5.39; N, 21.39; found: C, 64.02; H, 5.28; N, 21.20.

N^2^-(3-Phenylpropyl)-N^7^-(4-bromophenyl)-[1,2,4]triazolo [1,5-*a*]pyrimidine-2,7-diamine (**8u**)

Following general procedure C, the crude residue obtained by condensation of **11ad** and 4′-bromoaniline was purified by flash chromatography, using ethyl acetate as eluent, to furnish **8u** as a white solid. Yield: 82%, mp 150–152 °C. ^1^H-NMR (DMSO-*d_6_*) δ: 1.84–1.88 (m, 2H), 2.29 (s, 3H), 2.64 (t, *J* = 8.0 Hz, 2H), 3.26–3.29 (m, 2H), 6.18 (s, 1H), 6.64 (t, *J* = 8.0 Hz, 1H), 7.15 (t, *J* = 7.2 Hz, 1H), 7.20–7.28 (m, 4H), 7.35 (d, *J* = 8.8 Hz, 2H), 7.59 (d, *J* = 8.8 Hz, 2H), 9.50 (bs, 1H). ^13^C-NMR (DMSO-*d_6_*) δ: 24.91, 31.61, 33.08, 42.32, 88.92, 118.02, 126.13 (2C), 126.55 (2C), 128.77 (3C), 132.66 (2C), 137.24, 142.40, 144.20, 155.93, 162.07, 166.59. MS (ESI): [M]^+^ = 437.3, 439.3. Anal. calcd for C_21_H_21_BrN. C, 57.67; H, 4.84; N, 19.22; found: C, 57.51; H, 4.68; N, 19.10.

N^2^-(3-Phenylpropyl)-N^7^-(3-ethynylphenyl)-[1,2,4]triazolo [1,5-*a*]pyrimidine-2,7-diamine (**8v**)

Following general procedure C, the crude residue obtained by condensation of **11ad** and 3′-ethynylaniline was purified by flash chromatography, using ethyl acetate as eluent, to furnish **8v** as a gray solid. Yield: 74%, mp 202–204 °C. ^1^H-NMR (DMSO-*d_6_*) δ: 1.85–1.89 (m, 2H), 2.29 (s, 3H), 2.65 (t, *J* = 7.2 Hz, 2H), 3.26–3.30 (m, 2H), 4.25 (s, 1H), 6.15 (s, 1H), 6.62 (t, *J* = 6.4 Hz, 1H), 7.15 (t, *J* = 7.2 Hz, 1H), 7.20 (d, *J* = 6.8 Hz, 2H), 7.22–7.30 (m, 3H), 7.33–7.35 (m, 1H), 7.43–7.48 (m, 2H), 9.46 (bs, 1H). ^13^C-NMR (DMSO-*d_6_*) δ: 24.95, 31.62, 33.10, 42.35, 81.91, 83.35, 88.91, 123.23, 125.17, 126.12, 127.46, 128.72 (2C), 128.76 (2C), 129.18, 130.28, 138.16, 142.40, 144.32, 155.93, 162.07, 166.62. MS (ESI): [M+1]^+^ = 383.4. Anal. calcd for C_23_H_22_N_6_. C, 72.23; H, 5.80; N, 21.97; found: C, 72.08; H, 5.66; N, 21.78.

### 3.2. Biological Activity

#### 3.2.1. In Vitro Antiproliferative Activities

All derivatives characterized by the presence of a 3′,4′,5′-trimethoxyaniline moiety at the 7-position of the [1,2,4]triazolo [1,5-*a*]pyrimidine nucleus and reference compound CA-4 (**1a**) were screened for antiproliferative activity against five human cancer cell lines (Table 1): non-small cell lung carcinoma (A549), breast cancer (MDA-MB-231), cervix carcinoma (HeLa), colon adenocarcinoma (HT-29) and human T-leukemia (Jurkat) cells. CA-4 had single-digit nanomolar activity (IC_50′_s,1–5 nM) against three of the five lines, with A549 and HT-29 cells more resistant to CA-4, with IC_50_ values of 180 and 3100 nM, respectively. Three of the thirty derivatives, the 4′-chlorobenzylamino (**7n**), 4′-methylbenzylamino (**7p**) and 3′,4′-methylendioxybenzylamino (**7y**) analogues had IC_50_ values < 1 μM against all five lines, while compounds **7f**, **7o**, **7s**, **7w** and **7ad** had IC_50_ values < 1 μM against four of the five cell lines. Derivatives **7j**, **7l**–**m** and **7o** had potent, double-digit nM IC_50′_s against the MDA-MB-231 cell line but were less active (IC_50_ > 100 nM) against the other cell lines. None of the synthesized compounds was as active as CA-4 against MDA-MB-231, HeLa and Jurkat cells, while seventeen of the thirty compounds showed antiproliferative activity against HT-29 cells superior to that of CA-4, with derivative **7n** being the most potent, with an IC_50_ value of 54 nM.

In the series of benzylamino derivatives **7i–y**, two of the synthesized compounds, **7n** and **7p**, the *para*-chlorobenzylamino and *para*-methylbenzylamino derivatives, respectively, were significantly more active than the other compounds, with IC_50_ values of 31–323 nM (**7n**) and 60–427 nM (**7p**). All compounds with a *para*-substituted phenyl at the 2-position of the [1,2,4]triazolo [1,5-*a*]pyrimidine nucleus, corresponding to *p*-F, *p*-Cl and *p*-Me derivatives **7a**, **7b** and **7c**, respectively, had IC_50_ values below 2 µM, while the corresponding aniline derivatives **7e–g** showed increased antiproliferative activity. The activity of 2-anilino-[1,2,4]triazolo [1,5-*a*]pyrimidine derivatives **7d–h** was more pronounced against HeLa cells, with the *p*-chloroaniline analogue **7f** the most active compound, with an IC_50_ value of 92 nM.

Comparing the activities of unsubstituted phenyl derivatives **7i**, **7z** and **7ad**, with the exception of the Jurkat cell line, antiproliferative activity increased as the length of the alkyl chain between the aryl moiety and the nitrogen at the 2-position of the [1,2,4]triazolo [1,5-*a*]pyrimidine ring grew from one (**7i**) to two (**7z**) to three (**7ad**) methylene units, with the 3-phenylpropylamino derivative **7ad** the most potent compound of the three, with IC_50_ values of 190, 91, 67 and 142 nM against the A549, MDA-MB-231, HeLa and HT-29 cell lines, respectively, but with a >2 µM IC_50_ obtained against the Jurkat cells.

Bioisosteric replacement of phenyl with 4′-pyridinyl in the benzylamino derivative **7i**, yielding **7j**, was detrimental for activity against A549 and Jurkat cells. The two compounds were equipotent against HeLa and HT-29 cells, but **7j** was 46-fold more active than **7i** against MDA-MB-231 cells (IC_50_, 59 nM). For the *p*-fluoroaniline derivative **7e**, the corresponding *p*-fluorobenzylamino homologue **7l** was 2-6-fold less active in four lines, but 12-fold more potent in MDA-MB-231 cells (IC_50_, 61 nM).

For the derivative **7l**, increasing the size of the halide from fluorine to chlorine, to yield the *p*-Cl-benzylamino analogue **7n**, produced a 17–60-fold increase in antiproliferative activity against four of the five cell lines, with particular potency against HeLa and HT-29 cells, with IC_50′_s of 31 and 54 nM, respectively, while a 3-fold reduction in activity occurred with MDA-MB-231 cells. Moving the chlorine atom from the *para*- to the *meta*-position (compound **7o**) caused a 9-24-fold reduction in activity against four of the five cancer cell lines, with a 3-fold increase in potency only against the MDA-MB-231 cells (IC_50_, 60 nM).

For the *p*-toluidine derivative **7g**, the corresponding *p*-methylbenzylamino homologue **7p** yielded a compound that was 2–9-fold more potent against all cancer cell lines, with IC_50_ values of 60–420 nM, while moving the methyl group from the *para*- to the *meta*- or *ortho*-position (compounds **7q** and **7r**, respectively) led to a dramatic drop in potency. Compound **7p** was 2–4-fold less active than the chloro derivative **7n**, while **7p** and **7n** were equipotent against the Jurkat cells.

Replacement of the methyl in compound **7p** with a more electron-releasing methoxy group, to furnish the *p*-methoxybenzylamino derivative **7s**, produced a 2–5-fold reduction in antiproliferative activity against all five lines, with IC_50_ values ranging from 217 to 900 nM against four of the five cell lines and an IC_50_ of 1.3 μM on the MDA-MB-231 cells. Relative to the activity of benzylamino substituted derivative **7i**, for the corresponding 3′,4′-methylendioxybenzylamino derivative **7y**, a 6–28-fold increase in activity occurred with all five cell lines, with IC_50_ values ranging from 110 to 780 nM.

In the series of substituted benzylamino derivatives **7l–y**, the presence of the electron-withdrawing chlorine group at the *para*-position of the phenyl ring, to yield **7n**, was essential for potent antiproliferative activity against four of the five cancer cell lines, with only three compounds, corresponding to its isomeric *meta*-chloro derivative **7o** and the two *para-* and *meta*-fluoro isomeric analogues **7l** and **7m**, being more potent that **7n** against the MDA-MB-231 cells. Replacement of chlorine with stronger electron-withdrawing groups, such as CF_3_ or CN (for compounds **7w** or **7x**, respectively) or electron-releasing groups such as methyl or methoxy (derivatives **7p** and **7s**, respectively) was generally detrimental for antiproliferative activity, leading to a drop in potency.

For the most active compounds **7n**, **7p**, **7y** and **7ad**, we investigated effects on growth inhibition against the five cell lines of replacing the 3′,4′,5′-trimethoxyanilino moiety at the 7-position of the [1,2,4]triazolo [1,5-*a*]pyrimidine nucleus with different substituted anilines to furnish compounds **8a–v** (Table 2). With the exception of 3-phenylpropylamino derivatives **8q–v**, the greatest antiproliferative activity was observed when there was a 3′,4′,5′-trimethoxyanilino ring at the 7-position of the triazolopyrimidine scaffold.

Four of the six compounds **8q–v**, with a common 3-phenylpropylamino moiety at the 2-position of the triazolopyrimidine nucleus and different arylamines at its 7-position, specifically **8q** (*p*-F), **8r** (*m*-Cl, *p*-F), **8s** (*p*-Cl), and **8u** (*p*-Br), had antiproliferative activities greater than that of **7ad** against the A549, HeLa, HT-29 and Jurkat cell lines.

For the *p*-chlorobenzylamino derivatives **8a–c**, a comparison of the C-7 substituent effect revealed that compound **7n**, containing the 3′,4′,5′-trimethoxyanilino moiety, was more effective than the corresponding substituted aniline analogues **8a** (3′-Cl, 4′-F), **8b** (3′-Cl) and **8c** (3′-ethynyl), with the potency decreasing following the order 3′,4′,5′-trimethoxyaniline (**7n**) > 3′-chloro-4′-fluoroaniline (**8a**) > 3′-ethynylaniline (**8c**) >> 3′-chloroaniline (**8b**).

Compared to the *p*-methylbenzylamino derivative **7p**, when the 3′,4′,5′-trimethoxyaniline moiety was replaced with a 3′-chloro-4′-fluoroaniline group (**8d**), a 1.5–3-fold reduction in activity was observed against all cell lines. The reduction in activity was even more pronounced if the 3′-chloro-4′-fluoroaniline (**8d**) was replaced with a 3′-chloroaniline moiety (**8e**).

For the 3′,4′-methylendioxybenzylamino derivative **7y**, replacement of the 3′,4′,5′-trimethoxyanilino moiety by 3′-substituted anilines with substituents such as fluoro (**8h**), chloro (**8j**), bromo (**8m**), trifluoromethyl (**8o**) and ethynyl (**8p**) had a detrimental effect on activity against all cell lines. The same detrimental effect relative to **7y** also occurred with the 3′,4′-dichloroanilino and 4′-trifluoromethylanilino derivatives **8k** and **8n**, respectively, suggesting that the presence of a 3′,4′,5′-trimethoxyanilino moiety was essential for the greatest potency. Cell-growth-inhibitory activity against three of the five cancer cell lines was maintained by replacing the 3′,4′,5′-trimethoxyaniline moiety with a 3′-chloro-4′-fluoroaniline (**8f**) or 4′-halogen substituted anilines such as fluoro (**8g**), chloro (**8i**) or bromo (**8l)**, with a 3-4-fold increase in activity relative to **7ad** against Jurkat cells, while a 2–3-fold reduction in potency against MDA-MB-231 and HT-29 cells was observed for compounds **8f–g**, **8i** and **8l**.

For the anilino derivatives **8a–v**, a comparison of the C-7 substituent effect revealed that the greatest antiproliferative activity was observed when the 3′,4′,5′-trimethoxyanilino ring is replaced with a 3′-chloro-4′-fluoroaniline (**7ad** vs. **8r**, **7n** vs. **8a**, **7p** vs. **8d** and **7y** vs. **8f**). For the 3′-chloro-4′-fluoroanilino derivatives **8a**, **8d**, **8f** and **8r**, replacement of the 3′-chloro-4′-fluoroanilino by the 3′-chloroanilino moiety had a detrimental effect on activity against all cell lines (**8a** vs. **8b**, **8d** vs. **8e**, **8f** vs. **8j** and **8r** vs. **8t**), suggesting that the presence of the electron-withdrawing fluorine atom in the 3′-chloro-4′-fluoroanilino moiety was essential for the greatest potency and that electronic rather than steric factors account for the differing potencies of these compounds.

A review of the data in Table 1 and Table 2, from the point of view of compounds that yielded IC_50_ values below 100 nM, revealed that six compounds (**7f**, **7j**, **7l**, **7m**, **7o**, **7p**) had this potency in one cell line, two compounds (**7n**, **7ad**) in two cell lines, two compounds (**8q**, **8u**) and the CA-4 control in three cell line) and three compounds (**8r**, **8s**, **8v**) in four cell lines. This allowed us to narrow down the compounds of interest for further study.

#### 3.2.2. Effects of Compounds **7ad**, **8q** and **8r** in Non-Tumor Cells

Anti-cancer drugs should ideally be selective against malignant cells without inducing toxicity towards normal tissues. To obtain a preliminary indication of the cytotoxic potential of these derivatives in normal human cells, three of the most active compounds (**7ad**, **8q** and **8r**) were assayed in vitro against peripheral blood lymphocytes (PBLs) from healthy donors. The three compounds showed an IC_50_ greater than 10 μM, both in quiescent lymphocytes and in lymphocytes in an active phase of proliferation induced by phytohemagglutinin (PHA) as a mitogenic stimulus (Table 3). The reference compound CA-4 was nontoxic to quiescent PBLs but had a very low IC50 value on stimulated PBLs. These data indicate that **7ad**, **8q** and **8r** have better cancer-cell-selective-killing properties than CA-4 and an excellent therapeutic index, with the potential to reduce chemotherapy-related side effects.

#### 3.2.3. In Vitro Inhibition of Tubulin Polymerization and Colchicine Binding

To investigate if the growth inhibition of cancer cells by compounds characterized by the presence of the [1,2,4]triazolo [1,5-*a*]pyrimidine nucleus was associated with their binding to the colchicine site of tubulin, compounds **7n**, **7p**, **7y**, **7ad**, **8q–s** and **8u–v** were evaluated for their inhibitory effects on tubulin polymerization and on the binding of [^3^H]colchicine to tubulin (in the latter assay, tubulin was examined at a concentration of 0.5 μM, while compounds and colchicine were at 5 μM). For comparison, CA-4 was examined in contemporaneous experiments as a reference compound (Table 4).

Among the four compounds **7n**, **7p**, **7y** and **7ad** characterized by the presence of a common 3′,4′,5′-trimethoxyanilino moiety at the 7-position of the [1,2,4]triazolo [1,5-*a*]pyrimidine nucleus, derivatives **7n** and **7ad** strongly inhibited tubulin assembly and were almost twice as potent as CA-4 (IC_50_, 0.75 μM), although they were generally less potent than CA-4 as antiproliferative agents. The inhibitor potency on tubulin polymerization of **7p** was similar to that of CA-4, while derivative **7y** (IC_50_, 1.0 μM) was 1.5-fold less active than CA-4.

For the compounds **8q–s** and **8u–v**, with a common 3-phenylpropylamino and different substituted anilines at the 2- and 7-positions, respectively, of the [1,2,4]triazolo [1,5-*a*]pyrimidine nucleus, derivatives **8q, 8s** and **8u** inhibited tubulin assembly at the same level of activity as 3′,4′,5′-trimethoxyaniline derivative **7ad**, thus also being twice as potent as CA-4, while derivatives **8r** and **8v** were slightly more active with IC_50_ values of 0.52 and 0.43 μM, respectively. These data established that the trimethoxyphenyl moiety, a well-defined pharmacophore for the inhibition of tubulin polymerization found in colchicine, CA-4 and podophyllotoxin, was not essential for potent inhibition of tubulin polymerization in this group of compounds.

In the colchicine binding studies, compounds **8q–s** and **8u**, characterized by the presence of substituted anilines instead of a 3′,4′,5′-trimethoxyaniline at the 7-position of [1,2,4]triazolo [1,5-*a*]pyrimidine system, had quantitatively similar effects, varying within a narrow range (84–88% inhibition), showing potency comparable to that CA-4, which in these experiments inhibited colchicine binding by 97%. It is interesting to note that the 3′,4′,5′-trimethoxyaniline derivative **7ad** showed less binding affinity (55% inhibition) toward the colchicine binding site, although it showed potent inhibition of tubulin polymerization at the same levels as derivatives **8q**, **8s** and **8u**.

Derivatives **7n**, **7p**, **7y** and **7ad**, with a common 3′,4′,5′-trimethoxyaniline moiety at the 7-position of the [1,2,4]triazolo [1,5-*a*]pyrimidine nucleus, were less active than CA-4 at inhibiting the binding of [^3^H]colchicine to tubulin, with 19–55% inhibition.

It is significant that three compounds (**8q, 8s** and **8u**) had activities two-fold superior to that of CA-4 as inhibitors of tubulin assembly and potency similar to that of CA-4 as inhibitors of colchicine binding even though these compounds were less active as antiproliferative agents than CA-4 in three cell lines (MDA-MB-231, HeLa and Jurkat cells).

These results indicate that the antiproliferative activity of these compounds derives from an interaction with the colchicine site of tubulin and interference with microtubule assembly.

#### 3.2.4. Molecular Modeling Studies

The newly designed derivatives resemble the molecular conformation of the tubulin polymerization inhibitors colchicine and CA-4. Therefore, their potential ability to occupy the tubulin colchicine site has been evaluated by molecular docking.

The [1,2,4]triazolo [1,5-*a*]pyrimidine core of all derivatives lies in the central part of the active site, mimicking the binding of the co-crystallized colchicine (Figure 3). The 5-methyl group is placed at an optimal distance of 3.0–3.2 Å to interact with the βMet259, in a hydrophobic portion of the binding site where the colchicine methoxy group is positioned. The group at position 2 of the central core, regardless of its length, is sited at the interface between the two tubulin subunits, pointing toward a loop in the α-subunit (αSer178-αThr179) forming some anchoring interactions with the surrounding residues. The trimethoxyphenyl ring in position 7 is placed in proximity of βCys241, forming an important key interaction point for tubulin polymerization inhibition. According to this analysis, all these novel [1,2,4]triazolo [1,5-*a*]pyrimidine derivatives bearing at their 7-position a 3′,4′,5′-trimethoxyanilino moiety could bind this area. As a consequence, we can speculate that other factors different than the occupation of the active pocket, such as specific electrostatic/steric properties of both this tubulin region and the compounds, or physical properties specific for each molecule, could influence both the compound-protein interaction and the biological activity in the different cell-based assays, resulting in less active inhibitors. The reduced potency of several molecules on the panel of cancer cell lines can possibly be rationalized by a limited penetration into the cells or any other mechanism limiting the accessibility of these molecules to the cellular tubulin.

Interestingly, in the series of 3-phenylpropylamino derivatives **7ad** and **8q–v**, the replacement of the trimethoxyphenyl ring at the C-7 position of compound **7ad** with differently substituted phenyl rings does not affect the potential occupation of the colchicine site, being in line with the retained ability of these compounds to inhibit tubulin polymerization (Figure 4).

According to our previous findings [42,43], the presence of a dimethoxyphenyl or trimethoxyphenyl group is an essential structural feature for optimal occupation of the colchicine site, and the replacement with differently substituted phenyl rings has a detrimental effect on inhibition of tubulin polymerization as a potential consequence of a reduced affinity for the binding site. In this new series, the flexibility and the length of the 3-phenylpropylamino group at position 2 of compounds **8q–v** could favor an optimal occupation of the binding site, retaining the interaction with βCys241 and compensating for the absence of the methoxy groups.

#### 3.2.5. Compounds **7ad**, **8q** and **8r** Induced Alterations of the Microtubule Network

With the purpose of finding further evidence about the binding effect on tubulin of compounds **7ad**, **8q** and **8r**, we examined their ability to alter the cellular microtubule network by immunofluorescence analysis. As shown in Figure 5, the microtubule network exhibited normal arrangement and organization in HeLa cells in the absence of drug treatment. In contrast, after a 24 h treatment, with **8q** even at the lowest concentration used (10 nM), microtubules were disorganized, and the cells developed a spherical morphology. These changes were even more evident when the cells were treated with the compound at 50 nM. Moreover, the cells showed nuclear changes, and there were giant spherical cells with multiple nuclei (Figure 5).

The appearance of multinucleate cells could be the consequence of mitotic slippage, since it has been demonstrated that MTAs alter cell cycle progression and mitotic spindle checkpoints: cells with an altered microtubule system exit from mitosis without successfully separating the chromosomes. These cells continue to progress through the cell cycle entering into the G1 phase with a double chromosome number (4N) rather than the normal 2N [44].

#### 3.2.6. Compounds **7ad**, **8q** and **8r** Induced Cell Cycle Arrest in G2/M along with Alteration of Cell Cycle and DNA Damage Checkpoint Proteins

It has been widely described that inhibitors of tubulin polymerization may lead to an erroneous chromosome alignment and interfere with normal cell cycle progression, inducing cell cycle arrest in the mitotic phase [45]. In order to evaluate this typical effect of MTAs, alterations in cell cycle progression were analyzed by flow cytometry after a 24 h treatment with increasing concentrations of **7ad**, **8q** or **8r** in HeLa cells (Figure 6).

As expected, the three compounds caused a significant G2/M arrest in a concentration-dependent manner. This phenomenon began at 25 nM with **7ad** or **8r**, and it increased progressively until at 100 nM more than 70% of the cells were arrested in the G2/M phase. The effect was even more pronounced with compound **8q**, which induced arrest in G2/M of 80% of cells at 50 nM (Figure 6B).

We next studied the association between **8q**-induced G2/M arrest and alterations in the expression of proteins regulating the mitotic checkpoint. The cdc2/cyclin B complex regulates both mitotic entry and exit, and its activation is a multi-step process that starts with the binding of the regulatory subunit, cyclin B, to cdc2, thus forming the mitosis-promoting factor (MPF). MPF remains in an inactive form until dephosphorylation of cdc2 at Thr14/Tyr15 occurs. The cdc25C phosphatase, in the dephosphorylated form, is responsible for phosphate removal at Tyr15 and subsequent activation of the cdc2/cyclin B1 complex to promote mitotic entry. Activated MPF in turn phosphorylates cdc25C, and this contributes to the burst of cdc2/cyclin B activity that drives mitosis, creating an activation loop [46,47]. Microtubule-interfering agents impair microtubule dynamics, leading to the disruption of the mitotic spindle and blocking cell cycle progression at the transition from metaphase to anaphase. Damage of the mitotic spindle activates the mitotic spindle assembly checkpoint, and this inhibits activation of the anaphase-promoting complex (APC). Thus, proteolysis of cyclin B is prevented, causing an accumulation of cyclin B and cdc2 in the dephosphorylated (active) form [45].

Western blot analysis revealed that in HeLa cells after a 24 h of treatment with **8q** there was a significant reduction in cdc2 phosphorylation (Tyr15) (Figure 6C). Additionally, changes in the phosphorylation state of cdc25C and a slight increase in cyclin B expression were also observed.

The mitotic check point is regulated in response to DNA damage or replication blocks. DNA replication stress sites or single-stranded damage sites induce the recruitment of ATR, which becomes phosphorylated at Ser428, triggering the activation of a DNA damage signal cascade. ATR kinase activates CHK1 via phosphorylation at Ser345, which exerts its checkpoint mechanism in response to blocked DNA replication, principally by regulating the cdc25 family of phosphatases.

Indeed, CHK1 phosphorylates cdc25C at Ser216, enhancing the export of cdc25C from the nucleus to the cytoplasm. Thus, cdc25C cannot exert its activity on the cdc2/cyclin B complex; thus, the activation of cdc2 and the transition into mitosis is blocked [46,47]. Because of the involvement of ATR and CHK1 in the DNA damage response, we evaluated their phosphorylation status after **8q** treatment. We found an appreciable increase in ATR phosphorylation (Ser428) and of its downstream target CHK1 (S345) (Figure 6C). Thus, treatment with **8q** induced the activation of the DNA damage signaling response with consequent accumulation of cyclin B and blockage of cells in the G2/M phase.

#### 3.2.7. Effects of **7ad**, **8q** and **8r** Treatments on DNA Synthesis and Cell Proliferation

Growth and replication are features of all cells in all living organisms, but cancer cells undergo uncontrollable growth and division, invading surrounding healthy tissues in the body. Unlike normal cells, which stop dividing when they are close together thanks to contact inhibition, tumor cells evade controls limiting the growth of proliferating systems. For this reason, one of the main challenges of chemotherapy is to arrest uncontrolled cell proliferation, to prevent cancer from spreading in the organism.

With the goal of evaluating effects on cell proliferation and DNA synthesis following drug exposure, we performed the cytofluorimetric EdU proliferation assay. After a 24 h treatment with **7ad**, **8q** or **8r** at 25 or 50 nM, HeLa cells were incubated with EdU for 3 h. Incorporated EdU labels newly synthesized DNA and, thus, could be detected by flow cytometry applying a “click” chemistry reaction, as described in the Experimental Procedures section.

As shown in Figure 6D, drug treatments inhibited proliferation in a concentration-dependent manner, confirming that **7ad**, **8q** and **8r** are effective in reducing HeLa cell proliferation. The effect was greatest after **8q** exposure, while **8r** activity was significant only at the highest concentration (50 nM) used. These data agreed with results described above and confirm that cell cycle arrest results in a reduced proliferation rate.

#### 3.2.8. Compounds **7ad**, **8q** and **8r** Induced Apoptosis in HeLa Cells

With the aim of evaluating the mechanism of cell death induced by compounds **7ad**, **8q** and **8r**, we performed a bi-parametric cytofluorimetric assay using annexin-V-FITC and PI, which stain phosphatidylserine and DNA, respectively, as described in the Materials and Methods section. We found that the three compounds induced apoptosis in a time- and concentration-dependent manner, starting from a 50 nM concentration (Figure 7). The extent of apoptosis induction observed with 50 nM **8q** was similar to what we observed with the other two compounds at 100 nM, in good agreement with the cell viability assay that showed that **8q** had one of the lowest IC50 values for HeLa cells among the compounds studied here.

#### 3.2.9. Apoptosis Induced by Compounds **7ad**, **8q** or **8r** Follows the Mitochondrial Pathway

Since many antimitotic derivatives induce apoptosis following the mitochondrial pathway [48,49,50,51], we evaluated whether **7ad**, **8q** and **8r** induced mitochondrial depolarization. We analyzed HeLa cells treated with the three compounds at 10, 50 or 100 nM for 24 or 48 h. The decrease in mitochondrial potential was evaluated by flow cytometry using the fluorescent probe JC-1 and represented in graphs as percentage of monomeric JC-1 (Figure 8A).

All three compounds induced mitochondrial depolarization in a time- and concentration-dependent fashion, as shown in Figure 8B. At 24 h with all compounds, there was a significant increase in the percentage of monomeric JC-1, which became highly significant after 48 h, in good agreement with data obtained from the annexin-V/PI assay. The increase in the percentage of monomeric JC-1 with a dose-dependent trend indicated a progressive dissipation of the mitochondrial membrane potential (ΔΨ_mt_) that is associated with the appearance of apoptotic cells. The effect was most pronounced in HeLa cells exposed to **8q** after both 24 and 48 h treatments.

It is well known that mitochondrial membrane depolarization is closely related to mitochondrial production of ROS [52,53]. Thus, we evaluated whether ROS production increased after a 24 h treatment with **7ad**, **8q** or **8r** at 10, 50 or 100 nM. We detected cellular ROS generation using the dye 2,7-dichlorodihydrofluorescein diacetate (H_2_DCFDA), which is oxidized to DCF by cellular ROS. HeLa cells upon exposure to **7ad**, **8q** or **8r** significantly increased ROS production in a dose-dependent manner (Figure 8C), in comparison with ROS basal levels observed in vehicle-treated control cells. As in the previous studies, treatment with **8q** induced a dramatic increase at the 50 nM concentration, in agreement with the dissipation of the mitochondrial potential described above.

These results provided additional evidence that compounds **7ad**, **8q** and **8r** induced apoptosis of HeLa cells via the mitochondrial pathway.

#### 3.2.10. Compound **8q** Induces Caspase-3 Activation and PARP Cleavage and Causes a Decrease in the Expression of Antiapoptotic Proteins

Because many proteins are involved in the regulation of apoptotic mechanisms, we performed a Western blot analysis to evaluate the expression of some apoptosis-related proteins in HeLa cells following treatment with **8q**.

A crucial step in the apoptotic pathway consists of a cascade activation of proteolytic enzymes called caspases that are constitutively expressed in their inactive forms (zymogens). The apoptotic process starts with the stimulation of the initiator caspases, by proteolytic cleavage, and their conversion to the active effectors that cleave the pro-caspase-3 [54]. This event is crucial for the propagation of the apoptotic signal, as observed in several studies after exposure to many antimitotic agents [48,49,50,51]. Furthermore, the active caspase-3 is partially or totally responsible for the proteolytic cleavage of the DNA repair enzyme PARP. Indeed, PARP is a marker of cells undergoing apoptosis, and it is cleaved by caspase-3 from its full-length 116 kDa form to an inactive 85 kDa form [55]. To deepen our understanding of the mechanism of cell death induced by **8q**, we investigated the activation of pro-caspase-3 and the cleavage of PARP after a 24 h treatment.

As shown in Figure 8D, we observed at 50 nM **8q** the appearance of the cleaved fragment of the executioner caspase-3 and the consequent cleavage of its substrate PARP.

Among apoptosis-related proteins, we investigated BCL-2, an antiapoptotic protein that resides in the outer mitochondrial wall and plays a role in avoiding cytochrome C release, preventing the loss of mitochondrial membrane potential and avoiding caspase activation [54]. Many studies have demonstrated that activation of initiator pro-caspase-9 and pro-PARP causes a decrease in BCL-2 expression, thus indicating the occurrence of apoptosis [55]. In addition, we evaluated the expression of MCL-1, another antiapoptotic member of the BCL-2 family. In addition to its antiapoptotic function, studies have shown that MCL-1 promotes many pro-survival and pro-proliferative signaling pathways [56].

Immunoblot analysis showed that treatment with **8q** induced a partial reduction in BCL-2 and MCL-1 expression, but only (Figure 8) at the highest concentration (50 nM) examined. Altogether, these results provide evidence that **8q** induces a downregulation of both BCL-2 and MCL-1 to disable their anti-apoptotic functions and potentiate cell death.

#### 3.2.11. Compound **8q** Impairs Cell Migration in HeLa Cells

Metastasis is the major cause of morbidity and mortality for cancer patients. As demonstrated in many studies, microtubules play a fundamental role in cell motility and, therefore, also in the metastatic process. In order to evaluate the activity of **8q** in inhibiting cell motility and migration, a HeLa cell monolayer was scratched, and cells were allowed to migrate to restore the monolayer. As shown in Figure 9, sublethal doses (0.625–10 nM) of **8q** significantly arrested the movement of cells, preventing wound closure. In particular, cell migration was inhibited by treatment in a concentration- and time-dependent manner, thus suggesting an alteration of microtubule dynamics at the lower concentrations examined.

#### 3.2.12. Effects of **8q** Treatments on Zebrafish Embryos (Acute Toxicity Test)

To evaluate the acute toxicity of **8q**, we used the zebrafish embryo toxicity assay according to OECD guidelines [57]. We performed a preliminary toxicity evaluation to investigate whether the exposure to different concentrations of compound **8q** or CA-4, used as reference compound, were lethal for embryos or caused the emergence of phenotypic defects. Zebrafish larvae at 72 hpf were treated with two concentrations of **8q** (100 nM and 1 μM) or CA-4 (100 nM), and their development was monitored at 24 and 48 h time points.

As shown in Table 5, the zebrafish embryo toxicity screening assay revealed no death and no morphologic changes or developmental abnormalities (cardiac hypertrophy, cardiac edema or locomotion alterations) with **8q** for both experimental conditions, in comparison with the nontreated control (Figure 10). Absence of lethality was seen also at the higher investigated concentration (1 μM), which is 34 times the IC_50_ (29 nM) of **8q** in Hela cells, suggesting that **8q** was well tolerated by the embryos.

In contrast, embryos treated with CA-4 at 100 nM showed 100% lethality. These data were similar to the results obtained with PBLs from healthy donors and indicates that **8q** would be worth exploring in other in vivo models.

#### 3.2.13. In Vivo Antitumor Activity and Antimetastasis Effects of Compound **8q** in a Zebrafish Xenograft

To evaluate the effects of **8q** on tumor cell proliferation and metastasis in vivo, we generated a zebrafish xenograft model.

DiI positive-labeled HeLa cells were injected within the Duct of Cuvier of 48 hpf Tg (fli1: EGFP) zebrafish embryos. After xenotransplantation, embryos were treated with **8q** at 145 or 290 nM. After a 24 h incubation at 34 °C, we evaluated changes in fluorescence in the caudal area of each injected animal using a fluorescence microscope, to measure the effects of **8q** on tumor mass and tumor spread.

When tumor cells were microinjected into zebrafish embryos, the HeLa cells migrated into the tail region. As shown in Figure 11, in zebrafish embryos treated with **8q**, the number of circulating tumor cells was significantly reduced compared to DMSO-controls, in agreement with in vitro cytotoxic effects of **8q** on HeLa cancer cells.

Furthermore, as shown in Figure 11B, the fluorescence intensity decreased in a dose-dependent manner, confirming the effectiveness of **8q** in eradicating DiI-positive tumor cells (*p*-value 290 nM: 0.01; *p*-value 145 nM: 0.04).

## 4. Conclusions

We described the synthesis and data obtained with fifty-two compounds based on 3′,4′,5′-trimethoxyaniline (**7a–ad**) and variably substituted anilines (**8a–v**) at the 7-position of the 2-substituted-[1,2,4]triazolo [1,5-*a*]pyrimidine nucleus. Thirty-six of the fifty-two synthesized compounds showed antiproliferative activity against human colon adenocarcinoma HT-29 cells superior to that of CA-4, with **7n**, **8q–s** and **8u–v** possessing the highest overall potency, with IC_50_ values lower than 100 nm (49–83 nM), comparable with the data reported in the literature for the most commonly used cytotoxic agents (oxaliplatin, raltitrexed and irinotecan) in the treatment of colorectal cancer (CRC) [58]. The typical resistance of the colorectal carcinoma cell line HT-29 against CA-4 and analogous compounds bearing a phenolic group (R=OH) on the B-ring was previously reported to be partly mediated by an overexpression of the MRP-1 and MRP-3 efflux transporter [59].

Thirty-six of the fifty-two synthesized compounds showed antiproliferative activity against HT-29 cells superior to that of CA-4, and we can speculate that these derivatives are also poor substrates for these efflux pumps. Seven of the synthesized compounds had the best antiproliferative activities against non-small cell lung carcinoma (A549) and, overall, were more (**8r**, **8s** and **8u**) or as active as CA-4 (**7u**, **7ad**, **8q** and **8v**), with **8u** the most active derivative (IC_50_: 48 nM), showing approximately four-fold improvement over CA-4. Compound **8u** was considerably more active than doxorubicin (IC_50_ 11.6 μM) in one study [60] and less potent than paclitaxel, docetaxel and cabazitaxel (IC_50_ 3.5, 19.5 and 15 nM, respectively) in other studies [61,62].

In this study, we demonstrated that replacing the 3′,4′,5′-trimethoxyanilino moiety at the 7-position of the 3-phenylpropylamino derivative **7ad** with a 4′-fluoroaniline (**8q**), 3′-chloro, 4′-fluoroanilino (**8r**), 4′-chloroanilino (**8s**) or 4′-bromoanilino (**8u**) moiety resulted in a new series of compounds characterized by potent antiproliferative activity (IC_50_ values 22–233 nM in the five tumor lines examined). Compounds **8q–s** and **8u** strongly inhibited the polymerization of tubulin, with IC_50_ values of 0.52, 0.39, 0.39 and 0.38 μM, respectively, and were thus twice as potent as CA-4. The same derivatives strongly inhibited [^3^H]colchicine binding to tubulin, and the potent inhibition observed was similar to that obtained with CA-4. These data underlined that, for this class of compounds, the 3,4,5-trimethoxyphenyl moiety, a common group shared by many colchicine site inhibitors, and which was present in **7ad**, was not essential for maintaining potent antiproliferative activities and an interaction with tubulin at the colchicine site. Importantly, experiments carried out to evaluate the cytotoxicity of the most active compounds on both quiescent and PHA-stimulated human lymphocytes showed that they exhibited no toxicity up to 10 µM.

Furthermore, a preliminary experiment, in which we evaluated the effects of treatment with **7ad**, **8q** or **8r**, suggested that the damage to microtubules caused morphological changes in cell shape and internal architecture, including the formation of giant cells and multinucleated cells. From Western blot analysis, we also observed that treatment of HeLa cells induced increased expression of cyclin B and caused the phosphorylation of cdc25C. The presence of replicative stress sites increased the expression of phosphorylated ATR with the consequent activation of its signaling pathway. The mitotic block caused by antimitotic agents is also associated with a reduction in cell proliferation, confirmed by EdU proliferation assay results. Additionally, as a consequence of G2/M arrest, cells underwent apoptosis. Western blot analysis, after treatment of HeLa cells with **8q**, demonstrated activation of caspase-3 with cleavage of PARP, a crucial event in the apoptotic cascade. Moreover, a reduction in the expression of anti-apoptotic proteins BCL-2 and MCL-1 indicated that **8q** was a potent inducer of apoptosis in the HeLa cell line. Apoptosis is associated with the loss of mitochondrial membrane potential and generation of ROS, and we demonstrated that this occurred by flow cytometric analysis. These findings confirmed the activation of the intrinsic pathway of apoptosis, which is a typical consequence of treatment of cancer cells with antimitotic agents.

Even more importantly, **8q** had substantial in vivo anticancer activity because it reduced HeLa cell growth in xenografts implanted in zebrafish embryos. This activity occurred at low doses of **8q** (145 nM and 290 nM), which caused no developmental toxicity.

Our findings demonstrated that **8q** is a promising new tubulin-binding agent that warrants further testing in preclinical in vivo cancer models because it could improve common anticancer therapies. In particular, it was determined that compound **8q** was effective in the inhibition of human cervix carcinoma (HeLa) cell growth and may be a potential compound for the treatment of cervix cancer.

## Data Availability

Data supporting the reported results are available on request from the corresponding authors.

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
