# Peer review of "Synthesis and Biological Evaluation of Highly Active 7-Anilino Triazolopyrimidines as Potent Antimicrotubule Agents"

_pharmaceutics, 2022, doi:10.3390/pharmaceutics14061191_

Round 1

Reviewer 1 Report

The manuscript entitled " Synthesis and Biological Evaluation of Highly Active 7-Anilino Triazolopyrimidines as Potent Antimicrotubule Agents" from Oliva et al involved the previous structural analyses in order to select the desire potential moieties with the bioactive activity for further synthesis of novel potential antimitotic agents.

The authors reported the complete synthesis of 52 molecules through a synthetic pathway that is a classic route to produce the group of molecules of interest. Additionally, they confirmed the molecules identity with their structural characterization, they included appropriate biological in vitro and in vivo assays according to the aim of this research (antiproliferative assay for several human cancer cell lines, cytotoxicity in human PBLs, effects on tubulin polymerization, effect on colchicine binding to tubulin, cell cycle analysis, measurement of apoptosis, measurement of mitochondrial membrane potential and ROS production, EdU cell proliferation assay, western blot, immunofluorescence labeling of microtubules, experiments on Zebrafish model, among others) and bioinformatic analyses (molecular modelling) to explain the obtained results.

The introduction is according to the developed topic of the manuscript, and it has updated bibliographical references to support the research.

Moreover, the manuscript is clear, organize, and focused on the topic that is of growing interest due to the potential biological applications in the field of medicinal chemistry.

Additionally, the information they described is supported with clear and logical images/figures/tables that summarize all the required data for the eight synthetized small molecules.

The methodological development is according to the main goal of the author’s research and included several in vitro and in vivo biological assays to confirm the hypothesis. It is important to note the complete analyses the authors performed to achieve the potential application of these synthetic molecules as tubulin inhibitors.

Furthermore, some typo mistakes should be corrected (in yellow, pdf file attached) such as:

  • The utilization of @ for concentrations (maybe these are typo mistakes)
  • Table 1 and Table 2: the results expressed in nM should be in uM (5800 nm is better expressed in uM concentration…)

Although, I suggest improving the conclusions and future perspectives mentioning (venturing thoughts) which will be the possible applications of potential formulations which include these selected and bioactive molecules.

Finally, I would like to invite the authors to add the abbreviation list of words at the end of this manuscript.

I recommend the acceptance of this manuscript after the authors performed the suggested corrections/additions.

Reviewer 2 Report

I reviewed the manuscript entitled “ Synthesis and Biological Evaluation of Highly Active 7-Anilino 2 Triazolopyrimidines as Potent Antimicrotubule Agents” by Oliva, P. et al.

Point to be considered by authors:

The manuscript is excellent in description, in methodology and in discussion. 

The manuscript deals with the synthesis and characterizations of numerous antimitotic agents (52) and their efficiency in biological systems including in vivo

Q1. 1H NMR of 7ad should show the right integrations as it is wrote in the manuscript. It should appears at least for the compounds of interest 8j, 8k,8l..

Q2. Is there any explanation why the cytotoxic activity of the compounds tested is less for PBLPHA compared to CA-4

A discussion on which kind of ROS can me involved will be interested. Moreover, I am not sure that H2DCFA is a good probe to measure ROS when it is following the mitochondrial pathway. If Cytc is release it will oxidize H2DCFA…

Minor points

Measurement of ATP functions, OCR, ECAR and Oxidation of peroxiredoxin 1 and 3 could have been provided to show redox/stress.

  • Line 62, (?-tubulin)
  • Line 98 There is no compound 3b in the Figure 1
  • Line 98. however, binds and not however. binds
  • Line 111 10 ?M
  • 3,4,5-trimethoxyphenyl instead of 3,4,5-trimethoxylpheny
  • Line 168, Chemical shifts ( ?)
  • line 179, KMnO4 instead of KMnO4
  • Line 403. 10a-ad instead of 9a-ad
  • Line 456? 466 ?482 ? 1029
  • Line 699 (add space) ?
  • Line 801. 19.58. instead of 19.58,
  • Line 1314 H2DCFDA instead of H2DCFDA
  • Line 1405 and 1398. in vivo instead of in vivo

SI

  • Scheme 2. Reagents. a: ArNH2,, i-PrOH, reflux; b: NH2NH2.H2O, MeOH, rx, 18 h
  • IPrOH instead of MeOH ?
  • Scheme 3. Reagents. a: Appropriate ArCH2NH2 or Ar(CH2)2NH2 or C6H5(CH2)3NH2, i-PrOH, room temperature; b: NH2NH2.H2O,, MeOH, reflux, 18 h
  • isopropropanol

Reviewer 3 Report

The study shows promising selective antitumor agents derived from 7-anilino triazolopyrimidine, which induce apoptosis as the primary cell death pathway triggered by tubulin polymerization inhibition. The work has very interesting results, but the results section is very long and dilutes the main goal of the manuscript.  The authors should report only the significant results without the description of the assay or didactic comments. Instead, the authors should compare their findings with specific results from the literature on agents that produce similar effects to establish the advantages and disadvantages of their new molecules compared with the existing ones.  In my opinion, some weaknesses should be addressed to improve the quality of this manuscript.

Comments:

Important information is missing in the abstract, like the tumor cell lines, the antiproliferative concentration range obtained for each family (compounds 7a-ad and 8a-v), and the conclusion of the SAR analysis.
The symbol @ is found thorough the manuscript

Could the authors provide a hypothesis for the significative antiproliferative activity observed for compounds 7n, 7ad, 8q-s, and 8u-v compared with the other derivatives and CA-4? The reported IC50 is comparable with the effect of an effective antiproliferative agent for HT-29 cells?
What can the authors say about the antiproliferative activity of the same compounds now on A549 cell cultures? Are these results comparable with those found for other antiproliferative agents against these lung carcinoma cells? 
Besides the description of the antiproliferative activity of specific compounds in the tumor cell lines evaluated, in my opinion, it is needed a summary of the electron-donor and electron-withdrawing substituents at the 2- and 7-positions of triazolopyrimidine pharmacophore. The same summary should be performed for derivatives that include the anilino moiety (compounds 8a-v).
The above-suggested analysis allows for the identification of the structural and physicochemical properties of the most active antiproliferative agents.
Regarding molecular docking studies, specific interactions of compounds 7ad and 8q-v with the tubulin colchicine site should be identified including the bonding energy with key amino acids of the binding site. A comparison with the molecular interactions and binding energy should be performed considering colchicine and other specific inhibitors like CA-4. The compiled information could be very helpful to explain the experimental inhibition of tubulin polymerization described in section 3.2.3.
Figure 6B should include the statistical analysis compared with the control.
Figures 6C and 8D need to include relative intensity graphs with statistical analysis. 
The authors could suggest an explanation for the selective cytotoxicity observed, considering that both colchicine and CA-4 exhibit a remarkable antiproliferative activity and inhibit microtubule polymerization but also present side effects or non-favorable physicochemical properties. 

Conclusions should be modified according to the changes performed considering the above suggestions.

Round 2

Reviewer 3 Report

The authors addressed all comments satisfactorily. The manuscript is ready for publication in its current form.